# UnCLe: An Unlearning Framework for Continual Learning

## Abstract

Recent advances in deep learning require models to exhibit continual learning capability, allowing them to learn new tasks and progressively accumulate knowledge without forgetting old tasks. Concurrently, there are growing concerns and regulatory requirements to meet privacy and safety by discarding some knowledge through machine unlearning. With the rapidly rising relevance of continual learning and machine unlearning, we consider them together under a unified framework in this paper. However, the conflicting nature of past data unavailability arising from continual learning makes it challenging to perform unlearning with existing methods which assume data availability. Moreover, in the proposed setup, where tasks are repeatedly learned and unlearned in a sequence, it is another challenge to maintain the stability of the tasks that need to be retained. To address these challenges, we propose UnCLe, an Unlearning Framework for Continual Learning designed to learn tasks incrementally and unlearn tasks without access to past data. To perform data-free unlearning, UnCLe leverages hypernetworks in conjunction with an unlearning objective that seeks to selectively align task-specific parameters with noise. Our experiments on popular benchmarks demonstrate UnCLe's consistent unlearning completeness and ability to preserve task stability over long sequences.

## 1 Introduction

Progress in Artificial Intelligence necessitates models capable of continual updates throughout their lifespan. These updates can be either additive or reductive. Additive updates allow models to acquire new tasks over time, ideally without relying on past task data or disrupting knowledge from prior learning, which would result in the catastrophic forgetting of old tasks. This challenge of sequential learning while mitigating forgetting is addressed in the field of Continual Learning (CL) Wang et al. (2024). Conversely, reductive updates are achieved through Machine Unlearning, which focuses on selectively removing specific knowledge (Nguyen et al., 2022) to meet privacy or regulatory requirements. For instance, consider a diabetes detection model trained on data from multiple hospitals. If one hospital withdraws its data, selectively unlearning that subset is preferable to retraining the model from scratch. Modern deep learning models must support both additive and reductive updates, integrating CL and unlearning capabilities seamlessly.

Bringing unlearning capabilities into a continual learning system can provide other practical advantages besides data privacy. As a CL model accumulates tasks, its plasticity inevitably decreases, making it challenging to learn new tasks effectively Kirkpatrick et al. (2017). We hypothesize that unlearning can address model saturation in Continual Learning (CL) settings. In such scenarios, unlearning can restore flexibility by removing outdated or irrelevant knowledge. For example, consider a robot transitioning from an industrial environment to a household setting. With unlearning capabilities, the robot can discard obsolete industrial skills and acquire new skills suited to its household role. Reflecting these real-world demands, it is important to develop models and methods that continually learn and unlearn tasks to remain adaptive and efficient.

Data availability presents a key challenge in integrating unlearning into the CL setting. Existing unlearning methods Chundawat et al. (2023a); Foster et al. (2024a); Fan et al. (2024) often require access to the specific data that needs to be unlearned. This contrasts with the CL philosophy where data is discarded after training. Even when the data constraint is relaxed, and privacy concerns are

Figure 1: Diagram depicting how the model state changes as each request is processed. The model starts blank with zero expertise on any task. At each learning operation, the model gains expertise on that particular task, as represented by the colored chips added to the model state. Conversely, when a task is unlearned, the model loses expertise on that particular task, indicated through the removal of corresponding task chips.

set aside, permitting a memory buffer for past data, we find that repeated use of existing unlearning methods destabilizes the learned model. This instability can cause models to forget tasks they need to retain and, in some cases, struggle to learn new tasks effectively. The issue of instability extends past unlearning. We observe that with current unlearning methods, models recover their performance on unlearned tasks while learning new tasks. This unintentional recovery undermines the goal of unlearning, making it difficult to ensure that removed knowledge remains forgotten while still enabling effective learning of new tasks.

To tackle these challenges, we propose a unified framework for continual learning and unlearning that supports learning and unlearning tasks whenever required, as shown in Figure 1. Reflecting this philosophy, we introduce UnCLe: an Unlearning Framework for Continual Learning. UnCLe seamlessly integrates continual learning and unlearning through a hypernetwork-based architecture. This hypernetwork incrementally generates task-specific parameters for the main classifier, conditioned on task embeddings that are learned alongside the hypernetwork. For unlearning, UnCLe aligns the hypernetwork's output for the target task to white noise, requiring only the task embedding and no access to past data. A regularization term preserves the performance of retained tasks while ensuring complete unlearning of the target task. We evaluate UnCLe on popular benchmarks using diverse metrics to provide a comprehensive view of unlearning in a continual setting. Our experiments highlight UnCLe's superiority in unlearning, particularly in terms of efficacy and stability, compared to existing methods.

In summary, we make the following contributions:

1. We study unlearning in the context of continual learning and propose a problem setup that considers the restrictions arising from CL in performing unlearning.

2. We propose UnCLe, a framework designed for continual learning and data-free unlearning over a sequence of tasks.

3. We demonstrate UnCLe's unlearning efficacy in a continual setting and ability to learn new tasks better through a range of experimental setups, data sets, and metrics.

## 2 RELATED WORKS

Continual Learning (Wang et al., 2024) represents a class of methods that mitigate catastrophic forgetting and facilitate knowledge transfer between tasks. There are a myriad of ways in which this is achieved. Regularisation-based methods (Kirkpatrick et al., 2017; Li & Hoiem, 2017; von Oswald et al., 2020) harness a regularisation term in their objective that prevents interference from new tasks on parameters deemed important to older tasks. Replay-based methods (Rolnick et al., 2019; Riemer et al., 2019; Shin et al., 2017; Buzzega et al., 2020) utilize a memory buffer or use a generative model to replay samples from old tasks while training on new tasks. Parameter isolation methods (Mallya & Lazebnik, 2018; Yoon et al., 2018) divide existing parameters between tasks or grow the network by adding new parameters to accommodate new tasks without interference.

Unlearning methods can be categorized into three categories based on the requirement of data. Methods such as (Graves et al., 2020; Chundawat et al., 2023a; Cotogni et al., 2024; Golatkar et al., 2020; Kurmanji et al., 2023; Fan et al., 2024; Foster et al., 2024b) required both forget set data and retain set data to perform unlearning on the model. Apart from this, Foster et al. (2024a) is a method that

requires access to forget set only to perform unlearning. Chundawat et al. (2023b) proposes two different methods that require neither forget set data nor retrain set data to perform unlearning.

There are limited works that explore data-free unlearning in the context of continual learning such as (Shibata et al., 2021a) that leverages natural catastrophic forgetting to unlearn by ceasing to regularise on the forget-task, (Liu et al., 2022) that utilizes parameter isolation to perform learning and unlearning, and (Chundawat et al., 2023b) that uses a generative model to create the forget-set on the fly. As observed in the experiments, these existing methods have massive space utilization, poor efficiency, and are ineffective for intermittent continual learning and unlearning tasks.

# 3 BACKGROUND

## 3.1 CONTINUAL LEARNING

Continual Learning Wang et al. (2024) considers a problem setting wherein the model encounters a sequence of requests over time to learn tasks represented by task identifier $T_t$ and corresponding data set $D_t$, $\mathbf{R} = \{R_t\}_{t=1}^{|\mathbf{R}|} = \{(T_t, D_t)\}_{t=1}^{|\mathbf{R}|}$. These tasks have non-identical data distributions: $D_i \neq D_j, \forall i, j; i \neq j$. Traditional algorithms falter in such non-stationary settings and exhibit catastrophic forgetting of earlier tasks due to interference from new tasks Kirkpatrick et al. (2017). The prime directive of the CL paradigm is to mitigate catastrophic forgetting and facilitate inter-task knowledge transfer.

## 3.2 MACHINE UNLEARNING

Machine Unlearning (Nguyen et al.) is concerned with removing the influence of a selective subset of data on a trained model so that the model behaves as if it has never been trained on that data. Consider a model $A_l(D)$ trained on the dataset $D$ with the learning algorithm $A_l$ and subsequently removed of the influence of a forget-set $D_f$ using an unlearning algorithm $A_u$. Let $D_r = D \setminus D_f$ be the retain-set and $\mathcal{H}'$ be the hypothesis space; we then define unlearning as:

$$Pr(A_l(D_r) \in \mathcal{H}') = Pr(A_u(A_l(D), D, D_f) \in \mathcal{H}') \tag{1}$$

Specifically, this is known as exact unlearning, which can either imply that the parameter or the output distribution of an unlearned model should be equal to that of a model trained solely on the retain set. This is non-trivial to achieve in practice, so most works (Chundawat et al., 2023a;b) focus on approximate unlearning, relaxing the condition to imply an output distribution that is effectively indistinguishable for most practical purposes.

# 4 PROBLEM FORMULATION

We consider the problem of *Continual Learning and Unlearning* (CLU) where the model incrementally encounters a sequence of requests $\mathbf{R} = \{R_t\}_{t=1}^{|\mathbf{R}|}$, as depicted in Figure 1. Each request is a triplet $R_t = (I_t, T_t, D_t)$ containing the instruction $I_t$, the task identifier $T_t$ and the corresponding dataset $D_t$. The instruction can either be to learn a new task or to unlearn a learned task. When $I_t = L$ the model learns the task $T_t$ from the associated dataset $D_t = \{x_t^i, y_t^i\}_{i=1}^{|D_t|}$. Here, $x_t^i \in \mathcal{X}$, the covariate space, and $y_t^i \in \mathcal{Y}$, the label space. In the case of unlearning, $I_t = U$ and $D_t = \{\}$ as the requirement is to perform data-free unlearning. Given the absence of data, Equation 1 is thus modified to adhere to the CLU setting:

$$Pr(A_l(D_{\leq t \setminus f}) \in \mathcal{H}') = Pr(A_u(A_l(D_{\leq t}), T_f) \in \mathcal{H}') \tag{2}$$

where $A_l(D_{\leq t})$ is a model continually trained on a sequence of datasets $D_{\leq t}$ and $A_l(D_{\leq t \setminus f})$ is a model trained on the same sequence barring the forget task $T_f$. Due to the inaccessibility of the forget-task's dataset $D_f$ as per the CLU formulation, the forget-task's identifier $T_f$ takes its place. The aforementioned exact unlearning formulation is non-trivial to achieve in practice; hence, as with most unlearning methods, UnCLe is an approximate unlearning framework.

# 5 METHODOLOGY

The objective is to address the challenges of catastrophic forgetting, model instability, and data unavailability that stem from the proposed CLU setting. With this in mind, we consider a unified framework capable of both continual learning and unlearning over long sequences in the absence of any past data whatsoever. We propose an Unlearning Framework for Continual Learning (UnCLe) that leverages hypernetwork to perform continual learning over tasks and data-free unlearning.

A hypernetwork Ha et al. (2017) $\mathcal{H}(.\,;\phi)$ parameterized by $\phi$, is a meta-model (typically a neural network) that generates weights of the main network $\mathcal{C}(.\,;\theta)$ used to solve a task. In a CL setting von Oswald et al. (2020), the hypernetwork takes in a task embedding $e_t$ corresponding to some unique task $T_t$ and generates main network weights $\theta_t$ that best suit the data corresponding to the task under consideration. To prevent catastrophic forgetting, the hypernetwork is regularized so that previous tasks' generated parameters remain largely consistent. This is achieved through a knowledge distillation-inspired objective that minimizes the difference between the current hypernetwork output and that of a frozen hypernetwork

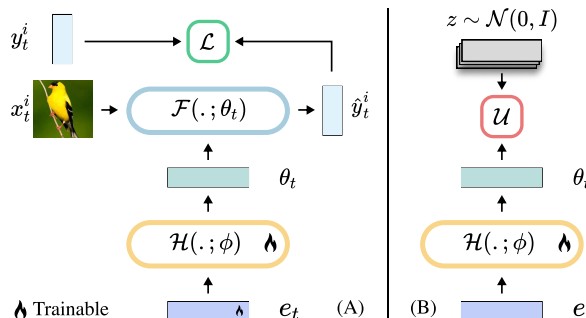

Figure 2: A schematic of the architecture showcasing the learning process *(A)* and the unlearning process *(B)*. Here, $\mathcal{L}$ represents the objective during learning as mentioned in Equation 3 and $\mathcal{U}$ represents the unlearning objective as mentioned in Equation 5.

copy made before learning the current task $t$. The use of embeddings makes for a negligible growth in parameters with each new task. Moreover, as the task-specific main network parameters are all generated by the hypernetwork, we also reap the benefits of inter-task knowledge transfer due to the sharing of hypernetwork parameters between tasks. The schematic of the architecture is presented in Figure 2.

In the CLU setting, when a learning request $R_t = (L, T_t, D_t)$ is encountered, the hypernetwork learns to generate main network parameters conditioned on task embedding $e_t$. The hypernetwork parameters $\phi$ and $e_t$ are learned by minimizing the task-specific loss ($\mathcal{L}_{task}$) computed using data set $D_t$. To prevent catastrophic forgetting over tasks that are retained (not considered for unlearning until $t$), a regularization term over those tasks enforces the hypernetwork to generate the same parameters for those tasks. This regularization considers knowledge distillation using a frozen hypernetwork with parameters $\phi^*$ prior to training on the current task. This results in the following learning objective: (here, $\beta$ controls the strength of the regularization.)

$$\underset{\phi, e_t}{\arg\min}\, \mathcal{L}_{task} + \beta \cdot \mathcal{L}_{reg}, \quad \text{where} \quad \mathcal{L}_{reg} = \frac{1}{t-1} \sum_{\substack{t'=1 \\ I_{t'} \neq U}}^{t-1} \|\mathcal{H}(e_{t'}; \phi^*) - \mathcal{H}(e_{t'}; \phi)\|_2^2 \qquad (3)$$

Conversely, for an unlearning request, $R_t = (U, T_f, \{\})$, task embeddings are the sole task-specific parameters in the hypernetwork framework. A trivial way to unlearn would be to discard the embedding $e_f$. But, as we have seen, parameter-sharing between tasks at the hypernetwork level enables inter-task knowledge transfer, inhibiting total unlearning of tasks. This can be seen empirically from Figure 3 (right) where we demonstrate that it is possible to partially recover the discarded task embedding by simply optimizing over a very small subset of samples and achieving performance that is very close to that of the original task embedding.

With unlearning, we require the model's prediction to be akin to a random guess, implying an output logit distribution close to a uniform distribution. A straightforward way to accomplish this in the model's output is to set all the model parameters to zero or, at the very least, the final layer parameters to zero. This leads to the model predicting 0 on all the logit nodes, the softmax of which is a uniform distribution. Based on this idea, we propose a new objective function to train the hypernetwork to more effectively unlearn the tasks to be unlearnt. To drive all the task-specific model parameters generated by the hypernetwork conditioned on task embedding $e_f$ to zero, we optimize the hypernetwork with the unlearning objective

$$\underset{\phi}{\arg\min}\, \gamma \cdot \|\mathcal{H}(e_f; \phi)\|_2^2 + \mathcal{L}_{reg}. \qquad (4)$$

To prevent unintended catastrophic forgetting on the other tasks, we utilize a similar regularisation formulation as with learning, controlled by the hyperparameter $\gamma$. Here, the regularisation skips out on the forgotten tasks and is calculated only over the tasks to be retained. Our experiments suggest

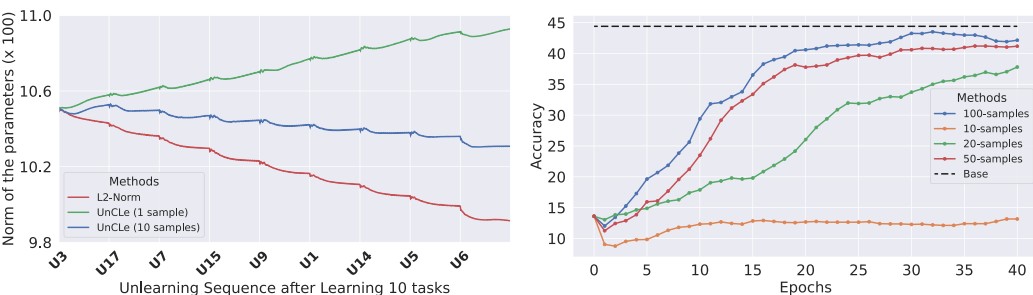

Figure 3: The plot on the left displays how the magnitude of hypernetwork parameters varies through the unlearning process for different noising strategies, comparing norm reduction and UnCLe's noise alignment with different values of noise sampling. The plot on the right demonstrates the recovery of task embeddings. Different lines denote different numbers of data samples used in recovery. "Base" denotes the accuracy obtained through the original embedding.

that while the norm reduction objective achieves unlearning with high $\gamma$ values, it comes at the cost of model stability, with the performance of the retained tasks and the ability to learn new tasks severely compromised. With this unlearning objective, our experiments indicate that lower $\gamma$ values are insufficient in attaining complete unlearning. We hypothesize that regressing the hypernetwork to generate zeros inadvertently drives some of the hypernetwork parameters acquiring values close to zero, which destabilizes the entire framework. Our empirical findings suggest that this is the case. Figure 3(left) clearly shows that the norm reduction objective consistently reduces the magnitude of the hypernetwork parameters with each unlearning task in the sequence.

To achieve a similar result without the destructive side effects, we approximate the $L^2$-norm term with a mean squared error (MSE) of the hypernetwork output with noise $z \sim \mathcal{N}(0, \mathbb{I}_d)$, averaged over $n$ samples. Adjusting $n$ allows us to control the severity of unlearning, leading to the following unlearning objective:

$$\arg\min_{\phi} \ \gamma \cdot \left( \frac{1}{n} \sum_{i=1}^{n} \|\mathcal{H}(e_f; \phi) - z_i\|_2^2 \right) + \mathcal{L}_{reg} \qquad (5)$$

We can understand the effect of $n$ on the unlearning process of driving hypernetwork output to zero from the theorems presented in Appendix A. We observe that as $n \to \infty$ MSE objective becomes similar to the norm reduction objective as shown in Lemma 1 in Appendix A). Theorem 3 in Appendix A) suggests that for a fixed deviation $\delta$, the probability that the average described in Equation 5 is far from the expected mean (squared norm over hypernetwork outputs) is inversely proportional to $n \times d$. In our case, as the dimension $d$ is that of a main network that is being generated by a hypernetwork, having a smaller value of $n$ can provide us sufficiently small probability. As portrayed in Figure 3(left), $n = 1$ is unstable in that it drives up the magnitude of the hypernetwork parameters, which could sometimes result in the accuracy of the retained tasks crashing. At the other end, $n = \infty$ is nothing but norm reduction that drives the hypernetwork parameters down to zero. The key is to strike a balance between the two extremes to achieve stable unlearning; thus, it needs to be chosen carefully. Algorithm 2 provides an unlearning algorithm for UnCLe.

The hypernetwork is optimized over the unlearning objective laid out in Equation 5 over a number of iterations that we term the burn-in. The burn-in can be tuned to suit the complexity of the model and data. Just as forward transfer enables quicker learning of successive tasks, we observed the same phenomenon with unlearning, where unlearning successive tasks became easier with each unlearning operation. We exploit this forward transfer in unlearning to improve overall unlearning efficiency by annealing the burn-in with each unlearning operation.

## 6 EXPERIMENTS

### 6.1 DATASETS

Our experiments are performed on the following datasets: (1) **Permuted-MNIST:** An MNIST variant with 10 random permutations of pixels as 10 different tasks proposed by Goodfellow et al.. (2)

**Algorithm 1** The Learning Algorithm

**Input**: Task $D_t$, Learning regularization constant $\beta$, Learning epochs $E_l$

```
 1: procedure LEARNING (D_t, β, E_l)
 2:     e_t = random_init()
 3:     for j = 0 to E_L do
 4:         for each batch (X_i, Y_i) in D_t do
 5:             θ_t = H(e_t; φ)
 6:             Ŷ_i = F(X; θ_t)
 7:             L_lrn = L_task(Y_i, Ŷ_i) + β · L_reg
 8:             Optimize {φ, e_t} w.r.t L_lrn
 9:         end for
10:     end for
11:     φ* = φ
12: end procedure
```

**Algorithm 2** The Unlearning Algorithm

**Input**: Forget Task Identifier $T_f$, Unlearning regularization constant $\gamma$, Burn-In $E_u$, Number of noise samples $n$

```
 1: procedure UNLEARNING(T_f, γ, E_u, n)
 2:     for j = 0 to E_u do
 3:         θ_f = H(e_f; φ)
 4:         L_fgt = 0
 5:         for k = 0 to n do
 6:             z ~ N(0, 𝕀_d)
 7:             L_fgt = L_fgt + (1/n)‖θ_f − z‖²₂
 8:         end for
 9:         L_ul = γ · L_fgt + L_reg
10:         Optimize {φ} w.r.t L_ul
11:     end for
12: end procedure
```

**5-Datasets:** A compilation comprising MNIST (Deng (2012)), Kuzushiji-MNIST (Clanuwat et al. (2018)), notMNIST (Bulatov (2011)), SVHN (Netzer et al. (2011)) and Fashion MNIST (Xiao et al.) forming five different tasks. (3) **CIFAR-100:** The CIFAR-100 dataset proposed by Krizhevsky, is divided into 10 different tasks. (4) **Tiny-ImageNet:** The Tiny-ImageNet dataset proposed by Moustafa, is divided into 20 different tasks. All datasets entail tasks with 10 classes each.

## 6.2 BASELINES

We compare UnCLe with existing methods that perform unlearning in a continual setting, namely **CLPU** (Liu et al., 2022) and **LWSF** (Shibata et al., 2021b). Due to the scarcity of works that perform unlearning in a continual setting, we adapt existing unlearning methods to the CL setting with a memory buffer of 500 samples per task as specified in the DER++ algorithm (Buzzega et al., 2020). In this manner, we consider the following baselines: **BadTeacher** by Chundawat et al., **SCRUB** by Kurmanji et al., **SalUn** by Fan et al. and **SSD** by Foster et al.. The aforementioned methods require data from both the retain and forget tasks during unlearning. We also consider **JiT** by Foster et al. that operates with just the forget-task data and **GKT** by Chundawat et al., which doesn't explicitly require memory buffers as it leverages a generative model to synthesize the required samples on the fly. In addition, we consider **JiT-Hnet** and **GKT-Hnet**, which utilize a hypernetwork for CL in DER++'s place. We also compare with standard baselines like fine-tuning (**FT**) and retraining (**RT** & **RT-Hnet**). FT and the two RT variants assume the availability of the complete retain-task data during unlearning. Further details related to baselines are provided in Appendix D.2.

## 6.3 METRICS

A CLU framework must be complete, efficient, stable, and undetectable. To measure each of these facets and paint a holistic picture of each unlearning method, we employ five diverse metrics: (1) **Retain-task accuracy (RA)** measures the average accuracy of the tasks that are retained at the end of the sequence. (2) **Forget-task accuracy (FA)** measures the average accuracy of the forgotten tasks. (3) **Unlearning Time (UT)** measures the average time taken to perform unlearning. (4) **Stability (SBY)** measures how well the algorithm preserves the stability of the learned tasks. (5) **Uniform (UNI)** measures the Jensen Shannon (JS) divergence of the output distribution of the model against a uniform distribution. It is scaled with $[1 − JS(.,.)] \times 100$ to match the other metrics. An ideal unlearning algorithm would have maximum RA, UNI, and SBY, minimum UT, and an FA of $\frac{1}{c}$ with $c$ being the number of classes for forget task, indicating an output that is as good as the random.

A unique problem that arises from unlearning in a continual setting is the instability of the learned model where the model's performance on retained tasks degrades with time due to multiple unlearning instances in sequence. To quantitatively measure this phenomenon, we present stability as a key metric of the CLU formulation, and is described in the following manner:

Consider $a_{ij}$ as the accuracy of the model on the $i^{th}$ task, evaluated after the $j^{th}$ request. Let $S_t$ be the stability of task $t$ and $S_{\mathbf{R}}$ be the overall stability of the request sequence. $S_t$ and $S_{\mathbf{R}}$ are thus computed:

$$S_{\mathbf{R}} = \frac{1}{T}\sum_{t=1}^{T} S_t, \quad \text{where} \quad S_t = \frac{1}{e-s}\sum_{k=s}^{e}\left[1 - \left(\frac{a_{ts} - a_{tk}}{a_{ts}}\right)\right] * 100 \tag{6}$$

Here, $T$ is the total number of tasks from the request sequence $\mathbf{R}$, and $s$ is the sequence index when a task $t$ is learned, and $e$ is the sequence index just before task $t$ is unlearned. Note that if a task is never unlearned in the request sequence, then $e$ would point to the final request on the sequence.

### 6.4 Implementation

We use a fully connected Hypernetwork with 3 hidden layers of dimensions 128, 256, and 512. The hypernetwork generates ResNet18 parameters in the case of Permuted MNIST experiments and ResNet50 elsewhere to demonstrate scalability. We defer ResNet18 results on other datasets to the E.4.1 of the Appendix. In the interest of efficiency, the parameters are generated in chunks, as obtaining them in a single pass would require a huge hypernetwork owing to the large ResNet sizes. Further information on the architecture of the hypernetwork and the chunking mechanism can be found in Appendix B. We use the Adam optimizer for both learning and unlearning, with a learning rate of 0.001. The learning rate is scheduled with a step learning rate scheduler with a step size of 25 epochs and a reduction factor of 0.5. We use a batch size of 512 across all the experiments. We train UnCLe with a different number of epochs for each dataset: 10 for Permuted-MNIST, 30 for 5-Datasets, 50 for CIFAR-100, and 100 for Tiny-ImageNet. Our models were trained on a single V100 GPU (32 GB).

### 6.5 Hyperparameter Tuning

While learning the hypernetwork, tuning $\beta$ plays an important role in balancing stability and plasticity. The values for $\beta$ were obtained through a search detailed in Appendix C.1. The chosen values for $\beta$ are as follows: 0.01 for Tiny-ImageNet, 0.001 for 5-Datasets, and 0.1 for both Permuted-MNIST and CIFAR-100.

The intensity of unlearning is controlled by two variables: the regularisation hyperparameter $\gamma$ and the burn-in period $E_u$. As with $\beta$ in learning, $\gamma$ balances the remembrance and the forgetting terms of the unlearning objective. The burn-in, $E_u$ controls the number of iterations the hypernetwork is optimised over the unlearning objective. A range of values for $\gamma$ and $E_u$ were explored as detailed in Appendix C.2. We use a burn-in of 100 iterations, annealed by 10% with each task, and a lower limit of 20 burn-in iterations. We use a $\gamma$ of 0.1 for 5 Datasets and 0.01 for the rest. For all the datasets, we used $n = 10$ noise samples.

## 7 Results

Our comparison of UnCLe with current unlearning methods is measured on five diverse metrics to paint a holistic picture of the unlearning process and highlight the strengths and shortfalls of each method. The results are benchmarked against a theoretical ideal unlearning framework, which would exhibit maximum accuracy on retained tasks (RA), maximum stability (SBY), a uniform output distribution (UNI), zero unlearning time (UT), and a forget-task accuracy (FA) of $\frac{1}{c}$, where $c$ is the number of classes. We conduct our experiments over 3 distinct sequences of operations, of which the first sequence's results are presented in Table 1, with additional results on the ResNet-18 backbone and other sequences (including mean and standard deviation) presented in Appendix E.5. To summarize and enable an intuitive visual comparison, Figure 5 portrays the performance signature of each framework on the Tiny-ImageNet test dataset across five metrics. Additional plots for other sequences are presented in Figure F.12.

UnCLe consistently outperforms all unlearning baselines on SBY and UNI metrics across all datasets, demonstrating its capability to excel in a continual learning setup. Moreover, UnCLe achieves FA values closest to a random prediction for all datasets, reflecting its thoroughness in unlearning. In terms of RA, UnCLe performs strongly and ranks among the top methods, even without relying on data for replay. Conversely, most baselines exhibit poor performance on retained tasks, as shown in Figure 4, but inflate their RA by relearning tasks from a memory buffer. As illustrated

| Methods | RS | FS | RA(↑) | FA(↓) | UNI(↑) | SBY(↑) | UT(↓) | RA(↑) | FA(↓) | UNI(↑) | SBY(↑) | UT(↓) |
|---|---|---|---|---|---|---|---|---|---|---|---|---|
| | | | Permuted-MNIST | | | | | CIFAR-100 | | | | |
| FT* | ✓ | | 94.47 | 67.70 | 19.93 | 98.52 | 1139 | **72.43** | 55.44 | 10.50 | 96.60 | 719.7 |
| RT* | ✓ | | 93.35 | 10.38 | 99.20 | 98.26 | 1532 | 62.91 | **9.690** | 99.19 | 92.45 | 577.4 |
| BadTeacher | ✓ | ✓ | 92.17 | 10.20 | 99.95 | 83.87 | 55.50 | 61.75 | 14.57 | 99.63 | 86.13 | 10.95 |
| SCRUB | ✓ | ✓ | 9.970 | 9.840 | -inf | 87.93 | 118.9 | 29.45 | 10.06 | -inf | 64.85 | 30.02 |
| SalUn | ✓ | ✓ | 92.39 | 59.24 | 98.47 | 93.53 | 358.3 | 66.56 | 44.89 | 59.85 | 89.32 | 51.47 |
| JiT | | ✓ | 86.93 | 29.90 | -3.76 | 84.52 | 213.7 | 65.94 | 43.93 | 22.11 | 87.31 | 24.01 |
| GKT | | | 89.77 | 12.13 | 96.64 | 72.46 | 36.08 | 57.05 | 10.70 | 95.97 | 70.23 | 68.61 |
| SSD | ✓ | ✓ | 86.32 | **9.930** | 99.66 | 71.88 | 35.16 | 43.27 | 10.00 | 99.97 | 65.95 | **5.730** |
| LwSF+ | | | 35.68 | 0.0 | 96.45 | 24.36 | — | 21.09 | 0.0 | 99.96 | 36.77 | — |
| CLPU | ✓ | | 91.73 | **0.0** | — | 97.22 | **0.0** | 63.10 | **0.0** | — | 91.44 | **0.0** |
| RT-Hnet* | ✓ | | 70.78 | 14.08 | -30.27 | 78.04 | 1685 | 23.81 | 9.710 | -1.240 | 63.53 | 784.9 |
| Hnet+ | | | 96.60 | 96.91 | -405.1 | 83.59 | — | 60.52 | 62.84 | -85.50 | 82.74 | — |
| Jit-Hnet | | ✓ | 76.81 | 10.27 | 89.58 | 76.51 | 257.5 | 60.79 | 16.97 | 74.97 | 85.20 | 22.94 |
| GKT-Hnet | | | 95.34 | 14.46 | 91.03 | 75.01 | 43.77 | 40.22 | 9.970 | 90.98 | 73.62 | 83.46 |
| UnCLe | | | **96.87** | 10.00 | **100.0** | **99.99** | 13.16 | 62.65 | 10.00 | **100.0** | **99.19** | 41.70 |
| | | | 5-Datasets | | | | | Tiny-ImageNet | | | | |
| FT* | ✓ | | 88.66 | 67.99 | 23.85 | 97.58 | 1592 | 60.08 | 52.56 | -11.47 | 95.55 | 694.2 |
| RT* | ✓ | | 84.79 | 9.600 | 99.76 | 96.58 | 1566 | 51.86 | 10.47 | 99.23 | 90.74 | 693.2 |
| BadTeacher | ✓ | ✓ | 54.38 | **8.550** | 99.99 | 86.14 | 76.78 | 52.79 | 15.73 | 99.55 | 83.76 | 8.680 |
| SCRUB | ✓ | ✓ | 9.160 | 12.97 | -inf | 77.55 | 171.1 | 19.48 | 10.00 | -inf | 71.13 | 32.52 |
| SalUn | ✓ | ✓ | 74.75 | 25.02 | 99.19 | 93.80 | 491.9 | **58.44** | 36.02 | 65.02 | 86.94 | 65.20 |
| JiT | | ✓ | 19.10 | 17.20 | -inf | 87.09 | 242.1 | 57.86 | 32.70 | 21.10 | 84.42 | 17.71 |
| GKT | | | 10.27 | 13.67 | 94.58 | 75.24 | 57.67 | 52.44 | 11.35 | 97.16 | 70.90 | 147.5 |
| SSD | ✓ | ✓ | 8.850 | 10.36 | 99.79 | 72.83 | 47.12 | 39.78 | 10.37 | 99.98 | 69.70 | **5.810** |
| LwSF+ | | | 31.76 | 0.0 | 99.98 | 51.21 | — | 17.58 | 0.0 | 99.97 | 35.28 | — |
| CLPU | ✓ | | 85.00 | 0.0 | — | 96.50 | **0.0** | 54.90 | 0.0 | — | 89.54 | **0.0** |
| RT-Hnet* | ✓ | | 76.23 | 18.44 | -108.5 | 95.63 | 1896 | 53.54 | **9.740** | -23.62 | 73.55 | 784.8 |
| Hnet+ | | | **94.56** | 96.73 | -380.9 | **99.99** | — | 57.53 | 54.31 | -72.66 | 76.06 | — |
| Jit-Hnet | | ✓ | 10.19 | 11.29 | -inf | 73.65 | 306.5 | 54.10 | 13.05 | 91.07 | 81.61 | 22.83 |
| GKT-Hnet | | | 10.53 | 14.48 | 88.66 | 77.19 | 83.34 | 44.40 | 9.850 | 94.43 | 73.61 | 75.75 |
| UnCLe | | | 94.12 | 10.04 | **100.0** | 99.91 | **33.28** | 55.24 | 10.00 | **100.0** | **98.19** | 29.63 |

Table 1: Table comparing UnCLe's performance with the baselines on each of the five metrics on 4 different datasets. Presented results are for Request Sequence 1 (Table D.6) averaged over 3 runs with different seeds. The table also highlights whether an algorithm requires access to the forget *(FS)* and the retain sets *(RS)* to perform unlearning. To enable comparison, the baselines have been augmented with a memory buffer to operate in a CL setting. '+' indicates methods that rely on catastrophic forgetting to enable unlearning. In such cases, metrics are calculated after the next learning request. '*' denotes methods that require the complete retain task data in unlearning. '−' indicates cases where metric calculation is not applicable.

in Figure 5, UnCLe achieves the greatest overlap with the ideal unlearning framework, excelling in SBY, FA, UNI, and achieving competitive RA and UT.

**Does unlearning protect data privacy?** To evaluate whether data is properly unlearned, we measure Membership Inference Attack (MIA) scores (Shokri et al., 2017). Our observations reveal that MIA scores are similar across most unlearning methods. Our setup employs different classifier heads for different tasks. When unlearning a task, the corresponding head is heavily randomized by all methods, resulting in indistinguishable representations. This randomization ensures equivalent MIA performance across all methods. A detailed explanation of MIA and accompanying results are deferred to Appendix D.3.4.

**Does UnCLe preserve stability?** Our experiments reveal critical shortcomings in baseline methods. We find that existing methods destabilize models when tasked with balancing the dual demands of preserving some tasks and unlearning others, as seen in Figure 4. Specifically, we observe that unlearning operations impact the performance of other learned tasks apart from the task to be unlearned. Our experiments show that such spillovers degrade RA. It is only due to replay during subsequent learning operations that the fallen RA is partially recovered. Similarly, we also find that once a task is unlearned, its accuracy almost recovers to where it was before the unlearning operation. This happens when other tasks are learned subsequently. UnCLe, however, maintains a stable

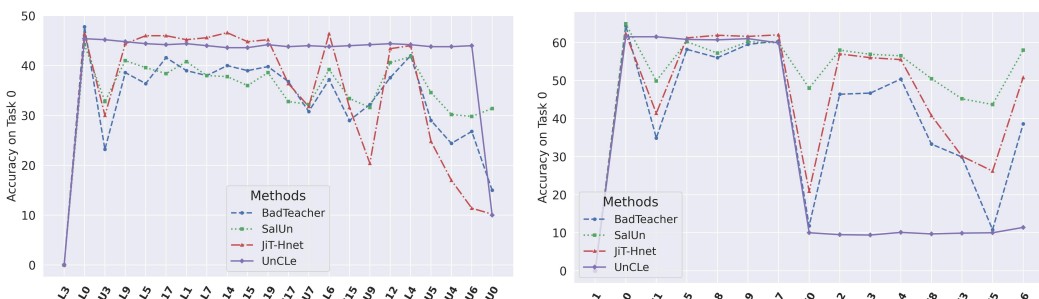

Figure 4: Plots tracking task 0's accuracy through a sequence of learning and unlearning operations on the TinyImageNet (left) and CIFAR 100 (right) datasets, comparing the stability of UnCLe with competing baselines

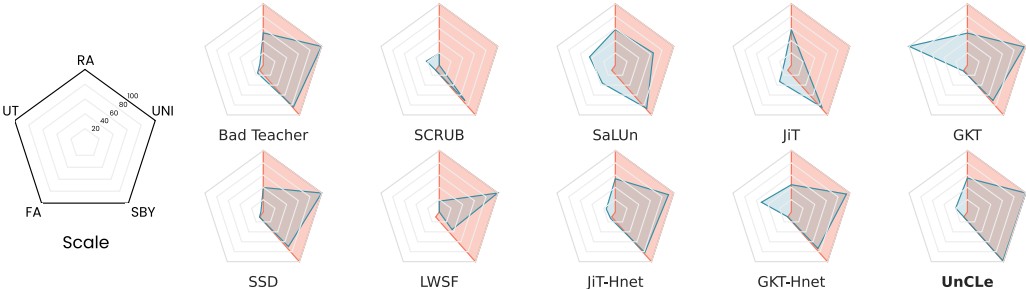

Figure 5: Radar plots comparing UnCLe's performance with the baselines on each of the five metrics on the Tiny-ImageNet dataset (Sequence 1). The pink plots display the signature of an expected ideal unlearning algorithm. The blue plots display UnCLe's performance and that of each baseline.

performance throughout the entire sequence of operations. We explore the implications behind these observations in great detail and include additional plots in Appendix E.4.

**Does unlearning help in learning future tasks better?** To answer this, we perform a comparison between a model that only learns tasks and UnCLe, which both learns and unlearns tasks. Table 3 displays the average RA obtained at the end of the operational sequence comparing both the cases. We observe that relinquishing unnecessary tasks provides tangible benefits, particularly with more complex datasets and longer sequences. In simpler dataset and sequences, we find minimal increase in performance, implying the model has not saturated. More detailed comparisons between UnCLe and Only Learning are presented in Appendix E.2. This highlights how unlearning not only serves as a privacy tool but also extends the longevity and maintainability of CL models by removing obsolete information.

| Methods | RA | FA | MIA | RA | FA | MIA |
|---|---|---|---|---|---|---|
| | **Permuted-MNIST** | | | **CIFAR-100** | | |
| **Fixed Noise** | 84.55 | **9.870** | 49.99 | 21.79 | 10.36 | 49.97 |
| **Norm Reduce** | 96.70 | 94.99 | 49.10 | **62.75** | 34.42 | 44.13 |
| **Discard $e_f$** | **96.87** | 61.79 | 49.11 | 60.21 | 20.7 | 46.88 |
| **UnCLe** | **96.87** | 10.00 | **50.00** | 62.65 | **10.00** | **50.00** |
| | **5-Datasets** | | | **Tiny-ImageNet** | | |
| **Fixed Noise** | 83.04 | 10.94 | 50.07 | 34.68 | **9.44** | 50.11 |
| **Norm Reduce** | 94.31 | 26.11 | 51.19 | 55.11 | 36.61 | 42.65 |
| **Discard $e_f$** | **94.52** | 80.91 | 50.25 | **56.50** | 15.54 | 48.44 |
| **UnCLe** | 94.12 | **10.04** | **50.01** | 55.24 | 10.00 | **50.01** |

Table 2: A comparison of UnCLe with alternative unlearning strategies.

| Methods | Permuted-MNIST | 5-Datasets |
|---|---|---|
| Only Learning | 96.84 | **94.12** |
| **UnCLe** | 96.87 | 94.12 |
| | **Tiny-ImageNet** | **CIFAR-100** |
| Only Learning | 50.47 | 60.51 |
| **UnCLe** | **55.24** | **62.65** |

Table 3: A comparison of average accuracy across the retained tasks from UnCLe versus a sequence with just learning tasks, demonstrating that unlearning old tasks helps learn new tasks better.

**How does UnCLe compare with other unlearning strategies?** We explore alternative unlearning strategies, such as Fixed Point Noising, Norm Reduction, and Discarding forget-task embeddings ($e_f$). A detailed comparison is presented in Table 2. Our findings indicate that UnCLe achieves better privacy by attaining near-perfect MIA scores. Additionally, UnCLe demonstrates holistic performance, excelling in RA and FA, while other strategies show significant weaknesses in one or more metrics. We defer further details and additional results to Appendix E.1

**Can the time taken to unlearn be reduced?** We exploit the forward transfer observed in unlearning to make UnCLe more efficient by annealing the burn-in iterations by 10%, with a lower limit of 20 iterations. Our experiments demonstrate that efficiency can be boosted thus without damage to unlearning efficacy. We defer further details and results to Appendix E.3.

**How are relevant tasks protected from unlearning spillover?** The term $\mathcal{L}_{reg}$ in Equation 5 serves to regularize the outputs of the current hypernetwork with that of the hypernetwork version before unlearning. This helps in keeping the effects of unlearning from spilling over to other tasks that need to be retained. Figure 6 presents the results of the ablation study, demonstrating the importance of $\mathcal{L}_{reg}$ during unlearning.

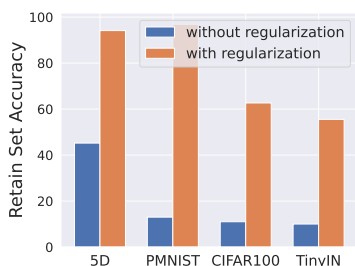

Figure 6: Plot provides a comparison of our approach UnCLe when used with and without regularization $\mathcal{L}_{reg}$ in the unlearning objective.

### 7.1 Key Takeaways

In building an Unlearning framework for Continual Learning, we identify a number of challenges: Firstly, current unlearning methods require past tasks' data to unlearn, which goes against the ethos of continual learning. UnCLe overcomes this data requirement through a novel hypernetwork-based solution and achieves **data-free unlearning**.

Secondly, stability remains a challenge for existing unlearning frameworks. Our experiments show that baseline methods destabilize the model when continually learning and unlearning. As seen in Figure 4 (left), existing unlearning methods unintentionally cause forgetting in tasks to be retained and rely on replay to help salvage lost performance during the next operation. On the other hand, UnCLe firmly maintains the **stability** of a task until it is unlearned.

A key expectation of any unlearning algorithm is to be thorough and permanent. Alarmingly, with existing unlearning methods, we find that on learning new tasks after unlearning, the performance of the unlearned task almost recovers to where it was before unlearning. We note that this troubling discovery should prompt future investigation. UnCLe achieves permanent unlearning where the unlearned tasks are **irrecoverable**. (Refer Figure 4 and Appendix E.4)

Finally, our experiments demonstrate by unlearning old and obsolete tasks, a model can **learn new tasks better**. This phenomenon is consistent across datasets, as presented in Table 3 and Appendix E.2.

## 8 Conclusion and Future Work

Recognizing the shortcomings of existing unlearning approaches in continual settings, we propose a unified treatment of continual learning and unlearning with UnCLe. Our experiments display UnCLe's effectiveness in addressing existing limitations such as model stability and unlearning completeness. Our experiments reveal that unlearning with existing methods is susceptible to recovery. We also show that unlearning obsolete old tasks helps learn future tasks better, opening new research avenues into more flexible CL frameworks. To address these, we proposed UnCLe, a novel hypernetwork-based data-free task unlearning framework that demonstrates stable unlearning performance, ensuring privacy protections and enabling greater continual learning flexibility.

UnCLe is capable of learning and unlearning tasks continually. However, UnCLe currently lacks the flexibility to learn and unlearn individual classes in a task in any arbitrary order. A future work is to imbue UnCLe with such granularity in learning and unlearning. Another future direction to study is how UnCLe can be applied to large pretrained transformer architectures that are continually fine-tuned on downstream tasks, either naively or through Parameter Efficient Fine Tuning methods (PEFT) such as LoRA, adapters, etc.

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

## APPENDIX

## A    CONNECTING MSE AND L2

**Lemma 1.** *Consider the parameters of a model to be $\theta \in \mathbb{R}^d$. A noisy approximation of the $L^2$-norm of parameters $\theta$ can be represented as an average of Mean Squared Error between parameters $\theta$ and samples $z_i \in \mathbb{R}^d$ from standard normal, $\mathcal{N}(0, \mathbb{I}_d)$. In other words,*

$$\lim_{n \to \infty} \frac{1}{n} \sum_{i=1}^n \|\theta - z_i\|_2^2 = \|\theta\|_2^2 + d$$

*Proof.* Consider $Y_i = \|\theta - z_i\|_2^2$ to be a random variable. Consider $E[.]$ as the function calculating the expectation of a random variable. As $z_i$ are i.i.d. samples of standard normal and $\theta$ is a constant, $Y_i$ are also i.i.d. samples. Using Strong Law of Large Numbers (Loève (1977)), we can say that:

$$\Pr\left[\lim_{n \to \infty} \frac{1}{n} \sum_{i=1}^n Y_i = E[Y_i]\right] = 1 \tag{7}$$

Now we would show that $E[Y_i] = \|\theta\|_2^2 + d$, where $d$ is the dimension of the parameter $\theta$.

$$\begin{aligned}
E[Y_i] &= E\left[\|\theta - z_i\|_2^2\right] \\
&= E\left[\theta^T \theta - z_i^T \theta - \theta^T z_i + z_i^T z_i\right] \\
&= E\left[\theta^T \theta\right] - 2E\left[z_i^T \theta\right] + E\left[z_i^T z_i\right] \\
&= \theta^T \theta - 2 \sum_j \theta_j E[z_{ij}] + \sum_j E[z_{ij}^2] \\
&= \|\theta\|_2^2 + \sum_j 1 \\
&= \|\theta\|_2^2 + d
\end{aligned} \tag{8}$$
$$\tag{9}$$
$$\tag{10}$$

Here, Equation 8 is using linearity property of expectation and Equation 9 uses the fact that $E[z_{ij}] = 0$ and $E[z_{ij}^2]$ is nothing but variance of that variable $z_{ij}$, which is equal to 1.

Based Equation 7 and Equation 10, we can say that,

$$\lim_{n \to \infty} \frac{1}{n} \sum_{i=1}^n \|\theta - z_i\|_2^2 = \|\theta\|_2^2 + d$$

$\square$

**Lemma 2.** *Bernstein's inequality (refer to Vershynin (2018)) Consider $X_1, X_2, \cdots, X_n$ as independent, mean-zero and sub-exponential random variables. Define $S_n = \sum_{i=1}^n X_i$. Then for every $\epsilon \geq 0$, we have*

$$\Pr\left[|S_n| \geq \epsilon\right] \leq 2 \exp\left[-c \min\left(\frac{\epsilon^2}{\sum_{i=1}^n \|X_i\|_{\psi_1}^2}, \frac{\epsilon}{\max_i \|X_i\|_{\psi_1}}\right)\right]$$

*where $c > 0$ is a constant and $\|.\|_{\psi_1}$ is 1-sub-exponential norm of a random variable.*

**Theorem 3.** *Consider $\forall i, Y_i \in \mathbb{R}^d$ is a random variable defined as $Y_i = \|\theta - z_i\|_2^2$, where $z_i \sim \mathcal{N}(0, \mathbb{I}_d)$. Define $S_n = \sum_{i=1}^n Y_i$. Then, with a relative deviation $\delta$,*

$$\Pr\left[S_n \geq (1 + \delta)E[S_n]\right] \leq \frac{2}{e^{\Theta\left(\min(\delta^2 nd, \delta n \sqrt{d})\right)}} \tag{11}$$

*where $\Theta(.)$ denotes the asymptotic average bound, commonly known as Big-Theta notation.*

*Proof.* We first need to understand the distribution of the random variable, $Y_i$. As $Y_i$ is the $L^2$-norm of a shifted $d$-dimensional standard normal distribution, $Y_i$ follows a *non-central chi-squared distribution* with $d$ degree of freedom and non-centrality parameter $\lambda = \|\theta\|_2^2$

$$Y_i \sim \chi_d^2(\lambda)$$

We know that the chi-square random variable is a sub-exponential random variable (Vershynin, 2018). We use the Lemma 2 to find the rate of convergence and its dependency with $n$ and $d$. To apply Lemma 2, we first need to centre the random variable,

$$X_i = Y_i - \mathrm{E}[Y_i] = \|\theta - z_i\|_2^2 - (\|\theta\|_2^2 + d) = \|\theta - z_i\|_2^2 - (\lambda + d) \tag{12}$$

Now, $X_i$ is a mean-zero sub-exponential random variable. Now, we need to compute the 1-sub-exponential norm of $X_i$. The chi-squared distribution is known to have a finite sub-exponential norm, but it's complex to compute, so we use an upper bound for it. Vershynin (2018) For a sub-exponential random variable with variance $\sigma^2$, sub-exponential norm satisfies, $\|X\|_{\psi_1} \le C\sigma$ where $C$ is some constant.

$$Var(X_i) = Var(\|\theta - z_i\|_2^2) = 2(d + 2\lambda) \tag{13}$$

As $\|\theta - z_i\|_2^2$ is a non-central chi-square distribution, we directly use its variance formula to get Equation 13. Now, for $X_i$, 1-sub-exponential norm is

$$\|X_i\|_{\psi_1} \le C\sqrt{2(d + 2\lambda)} \tag{14}$$

Applying Bernstein's inequality (Lemma 2) to $X_i$'s, we get,

$$\Pr\left[\left|\sum_{i=1}^n X_i\right| \ge \epsilon\right] \le 2\exp\left[-c\min\left(\frac{\epsilon^2}{2n(d + 2\lambda)}, \frac{\epsilon}{\sqrt{2(d + 2\lambda)}}\right)\right]$$

$$\Pr\left[\left|\sum_{i=1}^n (Y_i - \mathrm{E}[Y_i])\right| \ge \epsilon\right] \le 2\exp\left[-c\min\left(\frac{\epsilon^2}{2n(d + 2\lambda)}, \frac{\epsilon}{\sqrt{2(d + 2\lambda)}}\right)\right] \tag{15}$$

To analyze the upper tail bound, consider $S_n = \sum_{i=1}^n Y_i$.

$$\Pr\left[\sum_{i=1}^n (Y_i - \mathrm{E}[Y_i]) \ge \epsilon\right] = \Pr\left[S_n \ge \mathrm{E}[S_n] + \epsilon\right] \tag{16}$$

Let's define relative deviation $\delta$ as

$$\delta = \frac{\epsilon}{\mathrm{E}[S_n]} \ \Rightarrow\ \epsilon = \delta\mathrm{E}[S_n] \ \Rightarrow\ \epsilon = \delta n(d + \lambda) \tag{17}$$

Using Equation 15, Equation 16 and Equation 17 we can write that,

$$\Pr\left[S_n \ge (1 + \delta)\mathrm{E}[S_n]\right] \le 2\exp\left[-c\min\left(\frac{[\delta n(d + \lambda)]^2}{2n(d + 2\lambda)}, \frac{\delta n(d + \lambda)}{\sqrt{2(d + 2\lambda)}}\right)\right] \tag{18}$$

$$\le 2\exp\left[-c\min\left(\frac{\delta^2 n(d + \lambda)^2}{2(d + 2\lambda)}, \frac{\delta n(d + \lambda)}{\sqrt{2(d + 2\lambda)}}\right)\right] \tag{19}$$

Consider $\theta_j$ to be the value of $\theta$ on $j^{th}$ index. Then

$$\lambda = \|\theta\|_2^2 \ge d\theta_{min}^2 \quad \text{where} \quad \theta_{min} = \min_{1 \le j \le d} \theta_j$$

$$\lambda = \|\theta\|_2^2 \le d\theta_{max}^2 \quad \text{where} \quad \theta_{max} = \max_{1 \le j \le d} \theta_j$$

$$\frac{d^2(\theta_{min}^2 + 1)^2}{d(2\theta_{min}^2 + 1)} \le \frac{(d + \lambda)^2}{d + 2\lambda} \le \frac{d^2(\theta_{max}^2 + 1)^2}{d(2\theta_{max}^2 + 1)}$$

$$d\left(\frac{(\theta_{min}^2 + 1)^2}{2\theta_{min}^2 + 1}\right) \le \frac{(d + \lambda)^2}{d + 2\lambda} \le d\left(\frac{(\theta_{max}^2 + 1)^2}{(2\theta_{max}^2 + 1)}\right)$$

$$c_1.d \le \frac{(d + \lambda)^2}{d + 2\lambda} \le c_2.d \quad \text{where } c_1, c_2 > 0 \text{ are some constants}$$

$$\frac{(d + \lambda)^2}{d + 2\lambda} \approx \Theta(d)$$

Similarly, $\frac{d+\lambda}{\sqrt{d+2\lambda}} \approx \Theta(\sqrt{d})$

Based on the above claims, Equation 19 can rewritten as,

$$\Pr\left[S_n \geq (1+\delta)\mathrm{E}[S_n]\right] \leq 2\exp[-c.\Theta(\min(\delta^2 nd, \delta n\sqrt{d}))] \tag{20}$$

for some absolute constant $c > 0$. □

From the Theorem 3, we can observe that for a fixed deviation $\delta$, the probability that $S_n$ is far from $\mathrm{E}[S_n]$ is inversely proportional to $n \times d$.

## B  HYPERNETWORK

Hypernetworks $\mathcal{H}(.\,;\phi)$ are a class of neural networks designed to generate the parameters of another network, referred to as the target network $\mathcal{C}(.\,;\theta)$. Introduced by Ha et al. (2017), hypernetworks improve parameter efficiency and adaptability in machine learning models by learning a mapping from task-specific embeddings $e_t$ to the weights of the target network $\theta_t$, instead of directly optimizing the target network's weights. This enables greater flexibility in handling diverse tasks.

The hypernetwork framework comprises two main components:

1. **Hypernetwork**: A neural network responsible for generating the weights of the target network. In UnCLe, we employ a multi-layer perceptron as the hypernetwork.

2. **Target Network**: The primary network that performs the desired classification tasks using weights generated by the hypernetwork. Our experiments utilize both ResNet18 and ResNet50 as the target network.

When a learning request is encountered, the hypernetwork generates the main network parameters conditioned on the task embedding $e_t$. To achieve this, the hypernetwork parameters $\phi$ and the task embedding $e_t$ are optimized by minimizing the task-specific loss $\mathcal{L}_{task}$, which is computed using the data set $D_t$ corresponding to the current task. In our case, the task-specific loss is the Cross Entropy loss.

As tasks are learned continually, to ensure that knowledge of previously learned tasks is preserved, a regularization term is introduced. This term enforces the hypernetwork to generate consistent parameters for those tasks by aligning the output of the current hypernetwork with that of a frozen copy of the hypernetwork, denoted by $\phi^*$, saved prior to training on the current task. The regularization term leverages a knowledge distillation approach, comparing the outputs of the current and frozen hypernetworks for the embeddings of previous tasks.

The overall learning objective is defined as follows, where $\beta$ controls the strength of the regularization:

$$\underset{\phi,e_t}{\arg\min} \, \mathcal{L}_{task} + \beta \cdot \mathcal{L}_{reg}, \quad \text{where} \quad \mathcal{L}_{reg} = \frac{1}{t-1}\sum_{t'=1}^{t-1} \|\mathcal{H}(e_{t'};\phi^*) - \mathcal{H}(e_{t'};\phi)\|_2^2 \tag{21}$$

Here, $\mathcal{L}_{reg}$ represents the regularization term, calculated as the average squared difference between the outputs of the frozen and current hypernetworks for all previous tasks. This approach ensures that the parameters of the hypernetwork remain stable for previously learned tasks, effectively mitigating catastrophic forgetting.

They key benefit of using task embeddings to generate task-specific parameters results is negligible parameter growth as new tasks are added, ensuring high parameter efficiency. Since the hypernetwork generates all task-specific parameters and its core parameters are shared across tasks, it also facilitates inter-task knowledge transfer. This allows improvements in one task to benefit others.

A schematic representation of this architecture is presented in Figure B.7.

The hypernetwork consists of three hidden layers with dimensions 128, 256, and 512. Given the large size of the generated ResNet parameters, the hypernetwork's last layer becomes excessively

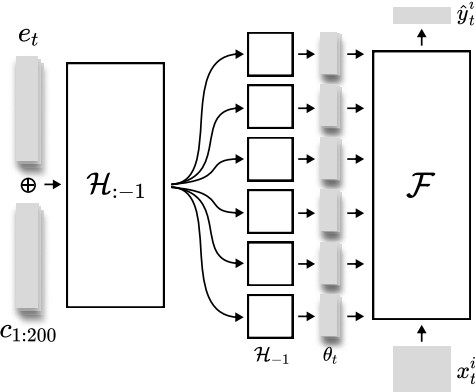

Figure B.7: Schematic of the architecture showcasing the task $e_{T_t}$ and chunk embeddings $c$, the hypernetwork and its various heads $\mathcal{H}$, the generated parameters $\theta$, the ResNet classifier $\mathcal{F}$ and, the input image $x_t^i$ and the predicted output $\hat{y}_t^i$.

large. To address this, we partition the main network parameters into smaller chunks and generate them separately. This significantly reduces the size of the hypernetwork's last layer, thereby minimizing the overall size of the hypernetwork.

Similar to how the hypernetwork generates task-specific networks by conditioning on unique task embeddings, it generates large networks in chunks by conditioning on unique chunk embeddings. These chunk embeddings are concatenated with task embeddings to create unique task-chunk embedding pairs, which generate the corresponding chunk of the parameters for the specific task network.

The chunk embeddings, like task embeddings, are learned through backpropagation. To prevent catastrophic forgetting, the chunk embeddings are frozen after the first task. In our implementation, both chunk and task embeddings have a dimension of 32. We found that dividing each task-specific network into 200 chunks strikes an effective balance between efficiency and performance.

Building on the previously described approach of generating task-specific network parameters in chunks, the hypernetwork further optimizes parameter generation by dividing its final layer into specialized heads. Each head is responsible for generating a specific type of parameter required for the target network: network weights, batch normalization parameters, and residual connection parameters. By explicitly separating the generation of different parameter types, the hypernetwork avoids generating unnecessary or redundant parameters. Each head is optimized to produce only the parameters relevant to its designated role, reducing computational overhead and memory usage.

The chunk-based parameter generation approach described earlier is seamlessly integrated with the specialized heads. For each chunk, the hypernetwork's heads produce only the subset of parameters required for that chunk, whether it is network weights, batch normalization parameters, or residual connection parameters. By generating parameters in chunks and assigning specialized roles to the final layer heads, the hypernetwork achieves a high degree of parameter efficiency. This design ensures that the size of the hypernetwork remains manageable even when generating large target networks like ResNet18 or ResNet50.

This architecture strikes an effective balance between scalability, modularity, and efficiency, making it well-suited for tasks requiring the generation of large and complex networks. The schematic of the hypernetwork used is described in Figure B.7.

## B.1 INITIALISATION

Classic weight initialization methods such as the Xavier Initialisation and the Kaiming He Initialisation, when applied on the hypernetwork, fail to generate classifier parameters in the correct scale. To counteract this, we employ Hyperfan Initialization, a principled parameter initialization technique

for hypernetworks proposed by Chang et al.. The goal of hyperfan initialization is to result in the generated parameters themselves following Kaiming He initialization.

## C  HYPERPARAMETER TUNING

### C.1  LEARNING HYPERPARAMETER: BETA

We perform a hyperparameter search to determine the best value for $\beta$. We perform experiments with $\beta$ values 1, 0.1, 0.01, and 0.001 and select the best-performing value for each dataset. The results of the hyperparameter search are presented in Table C.4:

| Dataset | 1 | 0.1 | 0.01 | 0.001 |
|---|---|---|---|---|
| Permuted MNIST | 96.24 | **96.68** | 96.64 | 96.52 |
| Five Datasets | 94.46 | 94.42 | 94.13 | **94.54** |
| CIFAR-100 | 48.58 | **72.16** | 52.62 | 15.72 |
| TinyImageNet | 34.33 | 35.74 | **53.7** | 48.49 |

Table C.4: Results of tuning hyperparameter $\beta$. The highest average accuracy values are highlighted in bold.

As apparent, the chosen values for $\beta$ are as follows: 1e-2 for TinyImageNet, 1e-3 for Five Datasets and 1e-1 for both Permuted MNIST and CIFAR-100.

### C.2  UNLEARNING HYPERPARAMETERS: GAMMA & BURN-IN

We perform a hyperparameter search to determine the ideal value for $\gamma$. Our search range comprises the $\gamma$ values 0.1, 0.01, and 0.001. Our selection of gamma is dependent on two factors, namely the Forget Set Accuracy (FA) and the Retain Set Accuracy (RA). A good unlearning algorithm should attain an FA of less than chance ($\frac{1}{c}$ where $c$ is the number of classes, in this case $10\%$). We first select all the $\gamma$ values that result in an FA $\leq 10$. We then pick the $\gamma$ that maximizes RA among those selected values. The results of the hyperparameter search are presented in Table C.5. We find that the burn-in of 100 is sufficient across datasets and we adopt it as standard in all our experiments.

| Dataset | FA | | | RA | | |
|---|---|---|---|---|---|---|
| | **0.1** | **0.01** | **0.001** | **0.1** | **0.01** | **0.001** |
| **Permuted-MNIST** | **10.412** | 10.417 | 17.907 | 96.524 | 96.544 | **96.602** |
| **CIFAR-100** | **8.000** | 10.830 | 17.190 | 70.950 | 71.817 | **72.173** |
| **5-Datasets** | 8.278 | **8.070** | 9.783 | **92.868** | 92.779 | 92.847 |
| **Tiny-ImageNet** | 10.000 | 10.000 | 10.000 | 45.590 | **48.625** | 48.623 |

Table C.5: Table depicting the FA and RA for various gamma values across datasets.

The chosen $\gamma$ values are 1e-1 for 5-Datasets and 1e-2 elsewhere.

## D  EXPERIMENTAL DETAILS

### D.1  OPERATION SEQUENCES

On each dataset, we perform experiments over three unique sequences of learning and unlearning requests generated through random seeds. Experiments on the Five Datasets benchmark are performed over sequences of 7 requests. For Permuted-MNIST and CIFAR-100 datasets, we utilize sequences of 15 requests, and for the Tiny-ImageNet dataset, we experiment with long 30-request sequences. The sequences used are presented in Table D.6.

| Datasets | Seq Nos | Sequences |
|---|---|---|
| **5-Datasets** **(7 requests)** | **1** | L0 → L1 → U0 → L2 → L3 → L4 → U1 |
| | **2** | L3 → L4 → L2 → L0 → L1 → U3 → U0 |
| | **3** | L0 → L2 → U0 → L4 → L3 → U2 → U4 |
| **Permuted-MNIST** **& CIFAR-100** **(15 requests)** | **1** | L1 → L0 → U1 → L5 → L8 → L9 → L7 → U0 → L2 → L3 → L4 → U8 → U3 → U5 → L6 |
| | **2** | L6 → L7 → L2 → L1 → L0 → U1 → L9 → U7 → U2 → U0 → L4 → U4 → L8 → U6 → L5 |
| | **3** | L7 → L1 → L2 → L8 → L0 → U1 → L3 → L6 → U3 → U2 → L4 → L5 → U8 → L9 → U7 |
| **Tiny-ImageNet** **(30 requests)** | **1** | L3 → L0 → U3 → L9 → L5 → L17 → L1 → L7 → L14 → L15 → L19 → U17 → U7 → → L6 → U15 → U9 → L12 → L4 → U5 → U4 → U6 → U0 → U1 → U14 → U12 → → L13 → L18 → L2 → L11 → L8 |
| | **2** | L12 → L13 → L5 → L8 → L2 → U8 → L14 → U13 → U5 → U2 → L3 → U3 → L16 → → U12 → L11 → U16 → L7 → L15 → L10 → L19 → L9 → U14 → U7 → L18 → L6 → → L1 → L0 → L4 → U6 → L17 |
| | **3** | L2 → L7 → U2 → L18 → L12 → U7 → U18 → L16 → L0 → U16 → U0 → L13 → L4 → → U12 → U13 → L9 → L19 → U19 → U4 → L10 → L14 → L5 → U5 → U10 → L11 → → L1 → U1 → L17 → L6 → L3 |

Table D.6: This table provides three different sequences that are used to understand the generalizability of our approach. Here, $L\#n$ implies 'learn task $n$' and $U\#n$ implies 'unlearn task $n$'. Also for different task we have different sequence length showing that our method can scale to longer sequences.

## D.2 BASELINES: ADDENDUM

Methods that use DER++ as the base CL method use a standard ResNet backbone architecture with independent heads for each task. As these are in a task incremental setting, we get task IDs during inference, which is used to choose the required head. For these methods, we used Adam Optimizer with a learning rate of $0.001$. For different datasets, learning epochs were different. 5-Datasets were trained for 20 epochs, CIFAR100 was trained for 30 epochs, Permuted-MNIST was trained for 10 epochs, and Tiny-ImageNet was trained for 30 epochs.

Methods that use Hypernetwork generate weights for the main network, which is a ResNet. In these cases the learning hyperparameters were the same as UnCLe.

**CLPU** Liu et al. (2022) is a method that perform exact unlearning. It requires apriori knowledge about which task has a possibility to be unlearned and which task will never be unlearned. Based on this information, the task that can be unlearned in the future is used to train an independent network. When they receive a request to unlearn a particular task, they just drop that network. In our CLU setup, no such assumption was made about the prior information, so we assume every task can get an unlearning request in the future. So, the direct implementation would be having independent networks for each task and throwing the network when an unlearning request is received. So, it is apparent that we will get a Forget set Accuracy of zero and an Unlearning Time of zero. Also, as the network is unavailable to us, we will not be able to calculate the divergence with uniform distribution.

**LWSF** Shibata et al. (2021b) introduces a new setup of learning with selective forgetting where, at every request, we will receive a set of classes to learn and a set of classes to forget. An extreme case of their setup is ours, where at every request, we either receive to learn classes or to unlearn classes. They introduced an approach using class-specific mnemonic codes. We observed that when their approach was applied to an extreme case like ours, they failed to unlearn the task. Their approach primarily used the advantage of learning and unlearning together and leveraged the catastrophic forgetting behavior of neural networks. So, to get the full potential of their approach, we calculated all the unlearning metrics for an unlearning operation after the next learning request arrives. Note that there can be multiple unlearning requests simultaneously; in that case, after all the unlearning, when the next learning comes, we will calculate all the unlearning metrics after that. As a reason for this modification, we don't compute unlearning time for this method as it won't be a fair comparison. For this method, we used a batch size of 200 with SGD optimizer and momentum as 0.9. We used a learning rate of 0.1 for all the datasets. For LWSF, Permuted-MNIST was not converging during training, so we didn't report results for this dataset.

**BadTeacher** Chundawat et al. (2023a) is a baseline that uses a random network as a teacher model for the forget set and uses KL-divergence to match the distribution of the forget class to that of a random model. For the retained set, it tries to reduce the cross entropy corresponding to the ground truth. For our CLU setup, we modified the algorithm where for the CL part, we use a DER++, experience reply-based method where the memory buffer is again used to get the retain and forget set. We performed a

**SalUn** Fan et al. (2024) targets specific model weights that are most influenced by the data to be removed (the data from forget set ) rather than modifying the entire model. This selective adjustment helps the unlearned model retain high performance on the remaining data. It needs to generate the weight saliency map corresponding to the forget set, which it does based on gradients. Based on this approach, we designed a baseline with DER++ as the base CL algorithm. To set this in a CL setup, the weight saliency mask needs to be created every time we encounter an unlearning request.

**SCRUB** Kurmanji et al. (2023) is designed to selectively remove knowledge of specific data points from a pre-trained model while maintaining overall model performance on the remaining data. Unlearning Phase (Forgetting): A student model is trained to deviate from the predictions of a pre-trained teacher model on the data that must be forgotten (the "forget set"). This step ensures that the model forgets specific information tied to those data points. Retention Phase: While the student model unlearns the forget set, it is simultaneously trained to match the performance of the teacher model on the remaining dataset (the "retain set"). This ensures that the model retains its predictive power on data that does not need to be forgotten. Based on this approach, we designed a baseline with DER++ as the base CL algorithm.

**SSD** Foster et al. (2024b) SSD operates as a two-step, post hoc method that does not require re-training the model, making it computationally efficient and suitable for scenarios where training data might not be readily accessible. Parameter Selection phase: SSD uses the Fisher information matrix to identify parameters crucial to the data that need to be forgotten. Dampening phase: It dampens these parameters' effects proportionally to their importance, allowing the model to forget the targeted data while maintaining performance on the remaining data. Based on this approach we designed a baseline with base CL algorithm as DER++.

**GKT & GKT-Hnet**: These baselines are based on the paper Chundawat et al. (2023b) where a generator is used to generate samples that are then used to forget information from the main network. We designed two methods, one that uses DER++ as the base CL algorithm and the other that uses Hypernetwork as the base algorithm.

**JiT & JiT-Hnet**: These baselines are based on Foster et al. (2024a), which leverages Lipschitz continuity to perform unlearning in a zero-shot manner. This approach involves smoothing the output of the model with respect to perturbations of the input data targeted for deletion, which helps in forgetting the specific data points while maintaining the model's overall performance. We used two different variants of this method for our setup, where one (JiT) uses DER++ as the base CL algorithm, and the other (JiT-Hnet) uses Hypernetwork as the base CL algorithm. We tuned the hyperparameters for each of these and found not much difference was achieved. So we have the same hyperparameters as provided in Foster et al. (2024a).

**Others** Apart from all these baselines, we also used **FT** where when an unlearning request is encountered, the current model is fine-tuned on the whole retain set. This also uses DER++ as the base CL approach. **RT** is one of the baselines that retrain the whole network from scratch on the retrain set to perform unlearning. **Hnet** is a baseline that uses a hypernetwork as the CL algorithm and uses the implicit forgetting nature of the neural network to perform unlearning. It just removes the the particular regularization for the forget task, so the unlearning will only be apparent once a new learning request is encountered. **RT-Hnet** is a baseline that uses Hypernetwork as the base CL algorithm, and whenever an unlearning request is encountered, it trains a new hypernetwork in a sequential fashion on the retrain set.

### D.3 Metrics: Addendum

#### D.3.1 Average Retain Set Accuracy

The Average Retain Set Accuracy (RA) measures *Unlearning Stability*, indicating undesirable spillover effects over the tasks to be retained. It is the mean of the accuracy of all the retained tasks measured at the end of the sequence.

#### D.3.2 Average Forget Set Accuracy

The Average Forget Set Accuracy (FA) is a measure of *Unlearning Completeness*. It is the mean of the accuracy of all the forget tasks, measured at the end of their respective unlearn operations. An ideal FA value should be close to $(100/N_c)$ where $N_c$ is the number of classes per task. All experiments performed with UnCLe entail tasks with 10 classes each, putting the ideal FA value at 10.

#### D.3.3 Output Divergence from Uniform Distribution

This is simultaneously a measure of *Unlearning Completeness* and *Unlearning Detectability*. An ideal unlearning algorithm should be both complete and undetectable in its wake. This metric measures the Jensen-Shannon divergence between the output logit distribution and the uniform distribution. An exact unlearning algorithm would report a divergence score of zero.

#### D.3.4 Membership Inference Attack

The Membership Inference Attack (MIA) metric is a critical tool in evaluating the effectiveness of machine unlearning methods. MIAs exploit the model's behavior to infer whether a specific data point was included in its training set, raising concerns about privacy and data retention. In the context of machine unlearning, the MIA metric is employed to measure how effectively a model has "forgotten" the training data. The objective is for the model to behave indistinguishably on forgotten data and new, unseen data, indicating successful unlearning. To evaluate this, adversarial attacks are used, where an attacker attempts to infer the membership status of data samples targeted for removal.

If a MIA value is 50%, it generally indicates that the attack performs no better than random guessing. In this context, the attack's ability to correctly determine whether a data point was part of the training set is equivalent to a coin flip, where the attacker has a 50% chance of correctly identifying membership or non-membership Tu et al. (2024). A 50% MIA value suggests that the model has successfully mitigated the attack, as the adversary cannot infer membership status with any meaningful accuracy.

| Datasets | 5-Datasets | | Permuted-MNIST | | CIFAR100 | | Tiny-ImageNet | |
|---|---|---|---|---|---|---|---|---|
| **Methods** | **Mean** | **Std** | **Mean** | **Std** | **Mean** | **Std** | **Mean** | **Std** |
| **FT*** | 49.56 | 0.22 | 49.63 | 0.07 | 45.00 | 0.66 | 45.26 | 0.73 |
| **RT*** | 49.95 | 0.37 | 49.98 | 0.07 | 49.82 | 0.50 | 49.72 | 0.23 |
| **BadTeacher** | 50.03 | 0.16 | 50.04 | 0.11 | 53.06 | 0.82 | 52.54 | 0.33 |
| **SCRUB** | 50.25 | 0.21 | 49.99 | 0.01 | **50.00** | 0.00 | **50.00** | 0.00 |
| **SalUn** | 50.25 | 0.29 | 49.85 | 0.13 | 46.26 | 0.42 | 47.47 | 0.73 |
| **JiT** | 49.99 | 0.17 | 49.95 | 0.08 | 45.80 | 0.73 | 47.28 | 0.15 |
| **GKT** | 50.05 | 0.08 | 49.99 | 0.01 | 49.88 | 0.20 | 49.93 | 0.06 |
| **SSD** | 49.98 | 0.03 | 50.01 | 0.01 | **50.00** | 0.00 | **50.00** | 0.00 |
| **RT-Hnet*** | 49.75 | 0.06 | 49.90 | 0.04 | 50.28 | 0.39 | 50.05 | 0.22 |
| **Jit-Hnet** | 50.10 | 0.06 | 50.02 | 0.08 | 48.74 | 1.11 | 49.39 | 0.24 |
| **GKT-Hnet** | 49.99 | 0.19 | 49.98 | 0.22 | 50.12 | 0.11 | 50.10 | 0.05 |
| **UnCLe** | **50.01** | 0.09 | **50.00** | 0.02 | **50.00** | 0.00 | **50.00** | 0.00 |

Table D.7: That table compares the MIA performance of different baseline approaches against UnCLe. Here, we provide results on all 4 datasets on request sequence 1, averaged across 3 seeds.

Table D.7 presents MIA values, with mean and standard deviation (std) across various methods and datasets such as Permuted-MNIST, CIFAR100, and Tiny-ImageNet. The values, which are around 50%, suggest a general trend where models are largely resistant to MIA, indicating that attackers have difficulty distinguishing between data points in and out of the training set.

As our setup is a setup for task unlearning with task incremental continual learning, we use different heads for different tasks. when forgetting a particular task, the corresponding head is severely randomized by each of the methods. So when performing MIA, the representation corresponding to the forget head is already random for all the cases, providing indistinguishable representations leading to an equivalent performance in MIA for all the methods.

Apart from this, our approach, UnCLe, exhibits near-perfect resistance to MIA, consistently showing a mean MIA value of 50.00% across all datasets. This means that the attacker's ability to infer whether a data point was part of the training set is equivalent to random guessing, signifying robust privacy protection.

# E   OTHER EXPERIMENTS

## E.1   BASELINES: ALTERNATIVE UNLEARNING STRATEGIES

| Methods | RA | FA | UNI | MIA | UT | RA | FA | UNI | MIA | UT |
|---|---|---|---|---|---|---|---|---|---|---|
| | **5-Datasets** | | | | | **CIFAR-100** | | | | |
| Fixed Noise | 83.04 | 10.94 | -inf | 50.07 | 18.74 | 21.79 | 10.36 | -inf | 49.97 | 25.76 |
| Norm Reduce | 94.31 | 26.11 | 52.44 | 51.19 | **18.3** | **62.75** | 34.42 | 41.27 | 44.13 | **25.39** |
| Discard $e_f$ | **94.52** | 80.91 | -214.0 | 50.25 | 0.00 | 60.21 | 20.70 | 11.21 | 46.88 | 0.00 |
| **UnCLe** | 94.12 | **10.04** | **100.0** | **50.01** | 33.28 | 62.65 | **10.00** | **100.0** | **50.00** | 41.70 |
| | **Permuted-MNIST** | | | | | **Tiny-ImageNet** | | | | |
| Fixed Noise | 84.55 | **9.870** | -inf | 49.99 | 10.48 | 34.68 | **9.440** | -inf | 50.11 | 22.62 |
| Norm Reduce | 96.70 | 94.99 | -49.56 | 49.10 | **10.34** | 55.11 | 36.61 | 0.80 | 42.65 | **22.42** |
| Discard $e_f$ | **96.87** | 61.79 | -64.54 | 49.11 | 0.00 | **56.50** | 15.54 | 6.88 | 48.44 | 0.00 |
| **UnCLe** | **96.87** | 10.00 | **100.0** | **50.00** | 13.16 | 55.24 | 10.00 | **100.0** | **50.00** | 29.63 |

Table E.8: Table exploring various noising strategies on each of the four datasets. Results are on Request Sequence 1. All the other unlearning hyperparameters ($\gamma$, $E_u$) are kept constant for these experiments.

We experiment with a variety of noising strategies and compare our approach to norm reduction and fixed noise perturbation. **Norm reduction** uses the unlearning objective from Equation 4. **Fixed noise perturbation** uses the objective $\|\mathcal{H}(e_f; \phi) - z\|_2^2 + \gamma \cdot \mathcal{L}_{reg}$ where the noise $z$ is fixed throughout all tasks. **Discard** $e_f$ is the baseline in which to perform unlearning, remove the forget task embedding $e_f$, and replace them with random embedding. From the Table E.8, we conclude that Fixed noise perturbation hampers the retain-task accuracy. We also observe that the forget-task accuracy it achieves, while lower than UnCLe in some instances, is marginally detectable, whereas UnCLe's output remains the closest to the uniform distribution. Norm reduction maintains good RA but exhibits poor unlearning. If further reduction in FA is attempted via increasing burn-in, it compromises the model's stability and impacts RA, as noted in the methodology. We also observe that UnCLe, compared to all the other baselines, has the closest MIA value to 50, proving its superiority in data privacy.

## E.2   SATURATION ALLEVIATION

A CL model is said to be saturated when the amount of free parameters available is insufficient to accommodate a new task without incurring catastrophic forgetting of old tasks. In the field of Continual Learning, saturation is typically encountered when a large number of tasks are learned relative to the model's size. Saturation is a prime motivation for dynamic architectures that can expand model capacity to accommodate a greater number of classes Yoon et al. (2018). However, dynamic architectures suffer from issues such as having a large memory footprint and little to no knowledge transfer.

A saturated model suffers from the stability-plasticity dilemma Kirkpatrick et al. (2017). Such a model loses all its plasticity owing to all its parameters being tasked with storing information pertaining to a large variety of tasks. Attempts to forcefully learn new tasks will compromise its stability, resulting in catastrophic forgetting of old tasks. In regularization-based CL, where the model capacity cannot be expanded, there is no existing solution that can enable the model to learn new tasks without compromising stability. In such situations, we hypothesize that unlearning can alleviate saturation by effectively removing old and obsolete tasks, thereby making way for new tasks.

The hypernetwork in UnCLe maintains separate task embeddings for each task. Each of these embeddings, when input into the hypernetwork, generates task-specific classifier models. The consistency of the generation as the model adds new tasks continually is preserved by a regularization term depicted in Figure 2. Whenever there is a new learning operation, the regularization term enforces that the output of the hypernetwork in its current state is similar to that of the hypernetwork before the current operation. To do this, a copy of the hypernetwork is made, and the copy's parameters are frozen. Now, as a new operation is performed and the hypernetwork's parameters change, the distillation-inspired regularization term makes sure that the hypernetwork's output for past tasks' embeddings remains consistent, thereby minimizing forgetting. As a task is unlearned, the hypernetwork is no longer regularized with respect to its embedding when it learns future tasks. As a result, this reduces the number of constraints on the hypernetwork, helping alleviate saturation and improving the learning of new tasks post-unlearning.

To empirically demonstrate this phenomenon, we perform a comparison between a model that only learns tasks and UnCLe, which both learns and unlearns tasks. We analyze the results in two ways. As presented in Algorithm E.8, we compare the performance of each task right after the learning operation. As we can observe, after every unlearning operation, there is a notable performance when the next task is learned compared to Only Learning. Furthermore, Figure E.9 compares the performance of the tasks that remain at the end of the sequence of operations. In both cases, we find that UnCLe consistently outperforms the baseline that only performs learning operations, demonstrating that unlearning old tasks help learn new tasks better.

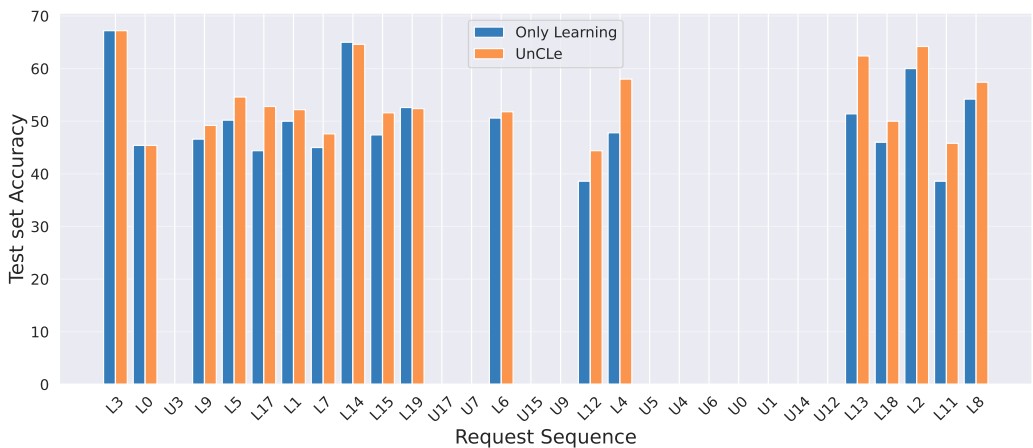

Figure E.8: A comparison between the individual task accuracies of UnCLe and a trivial baseline that only performs learning operations. Each of the above measurements are made immediately after the operation is performed. Note that tasks that follow unlearning operations consistently benefit from a higher accuracy. UnCLe outperforms the trivial baseline in every task that is retained.

### E.3   BURN-IN ANNEALING

We leverage the forward transfer observed in unlearning to enhance UnCLe's efficiency by introducing an annealing strategy for the burn-in phase. With each unlearning operation, the burn-in rate is reduced by 10%, with a minimum of 20 iterations to ensure stability. This progressive reduction capitalizes on the model's improved adaptability over time, significantly decreasing Unlearning

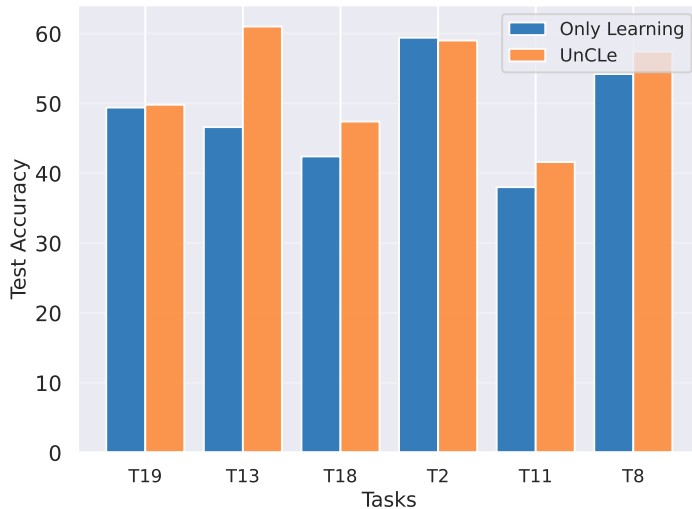

Figure E.9: A comparison between the final accuracies of the tasks that remain. UnCLe is compared with a trivial baseline that only performs learning operations. The measurements are made at the end of the sequence of operations.

Time (UT) without compromising performance. As shown in Table E.9, the Forget-Task Accuracy (FA) and Uniformity (UNI) metrics remain consistent, demonstrating that the annealing strategy maintains the quality of unlearning while optimizing computational efficiency.

| Methods | FA | UNI | UT | FA | UNI | UT |
|---|---|---|---|---|---|---|
| | CIFAR-100 | | | Tiny-ImageNet | | |
| without Annealing | 10.00 | 100.0 | 43.98 | 10.00 | 100.0 | 45.12 |
| with Annealing | 10.00 | 100.0 | **41.70** | 10.00 | 100.0 | **29.63** |

Table E.9: A comparison of UnCLe with and without Burn-In annealing.

## E.4 STABILITY

Stability remains a significant challenge for existing unlearning frameworks, particularly in scenarios involving the continual learning and unlearning of tasks. Our experiments reveal critical shortcomings in baseline methods, which tend to destabilize models when tasked with balancing the dual demands of preserving knowledge for some tasks while unlearning others. We present the results of our experiments in Figure 4 in the main paper and in Figure E.10 and Figure E.11 in the appendix. The instability of existing unlearning methods in continual settings manifests in two ways:

### E.4.1 FORGETTING RETAINED TASKS

Existing unlearning methods inadvertently cause catastrophic forgetting in tasks that are meant to be retained. This occurs because unlearning operations often modify shared model parameters, leading to unintentional degradation in the performance of previously learned tasks. Replay of data from previous tasks serves as the saving grace, helping salvage lost performance. However, this dependency on replay is not always practical, given that the lost performance will persist until a new learning operation follows. Even then, the lost performance almost never recovers fully.

We can observe this phenomenon in Figure E.10 and Figure E.11. In between the learning of the task and its eventual unlearning, we find that the task accuracy degrades whenever an unlearning operation is encountered only to rise back up when the next learning operation occurs. As mentioned, this is entirely due to replay, in the absence of which, the lost performance would remain lost. Various

baselines exhibit this instability in maintaining task accuracies to varying degrees whereas UnCLe stays close to the accuracy obtained right after the learning operation.

UnCLe, by contrast, is designed to maintain task stability firmly until a task is explicitly unlearned. This is achieved through the careful design of the hypernetwork and task-specific embeddings, which ensure that task representations remain untouched unless explicitly targeted for unlearning. This parameter isolation allows UnCLe to uphold the performance of retained tasks without requiring replay, making it a more efficient and reliable solution for continual learning and unlearning scenarios.

### E.4.2    REMEMBERING FORGOTTEN TASKS

A key expectation from any unlearning algorithm is that it must ensure unlearning is both thorough and permanent. Thoroughness implies that all knowledge related to the unlearned task is effectively erased from the model, leaving no residual influence on future operations. Permanence ensures that once a task is unlearned, its knowledge cannot be recovered when new tasks are introduced. Our findings highlight an alarming shortfall in existing unlearning methods: after unlearning a task, subsequent learning of new tasks can unintentionally restore the performance of the unlearned task to a level close to what it was before unlearning. As witnessed in Figure E.10 and Figure E.11, we find that the task accuracy jumps back up after unlearning when new tasks are learned. Various baselines exhibit this phenomenon to varying degrees whereas UnCLe stays close to the accuracy obtained right after the unlearning operation.

We believe that this occurs because existing methods often fail to completely eliminate the internal representations associated with the unlearned task. Instead, these representations may persist in latent forms within shared parameters or feature spaces, leading to unintended recovery when new tasks reinforce similar patterns. This troubling discovery raises serious concerns about the reliability and security of current unlearning frameworks, particularly in applications where permanent removal of knowledge is a regulatory or ethical necessity.

UnCLe directly addresses this issue by ensuring that unlearning is irreversible. Its hypernetwork-based architecture, coupled with a noise-alignment unlearning objective, thoroughly erases task-specific representations from the model. By aligning the outputs of the hypernetwork for unlearned tasks to noise, UnCLe effectively eliminates any trace of the unlearned task's influence on model behavior. Unlike existing methods, UnCLe prevents recovery of unlearned tasks when new tasks are subsequently introduced, making it a more reliable framework for permanent unlearning.

The stark contrast between UnCLe and existing methods underscores the importance of designing unlearning algorithms that meet the dual requirements of stability and permanence. The shortcomings of existing methods, particularly their inability to guarantee permanence, demand further investigation. Future work should focus on:

1. Analyzing Residual Representations: Understanding why and how unlearned tasks persist in shared model spaces and developing techniques to eliminate such residual traces.

2. Defining Robust Metrics: Establishing rigorous benchmarks and metrics for evaluating the thoroughness and permanence of unlearning beyond task-specific accuracy.

UnCLe's advancements in stability and permanence represent a significant step forward in continual learning and unlearning. By addressing critical challenges in a robust and efficient manner, it sets a strong foundation for the next generation of unlearning frameworks.

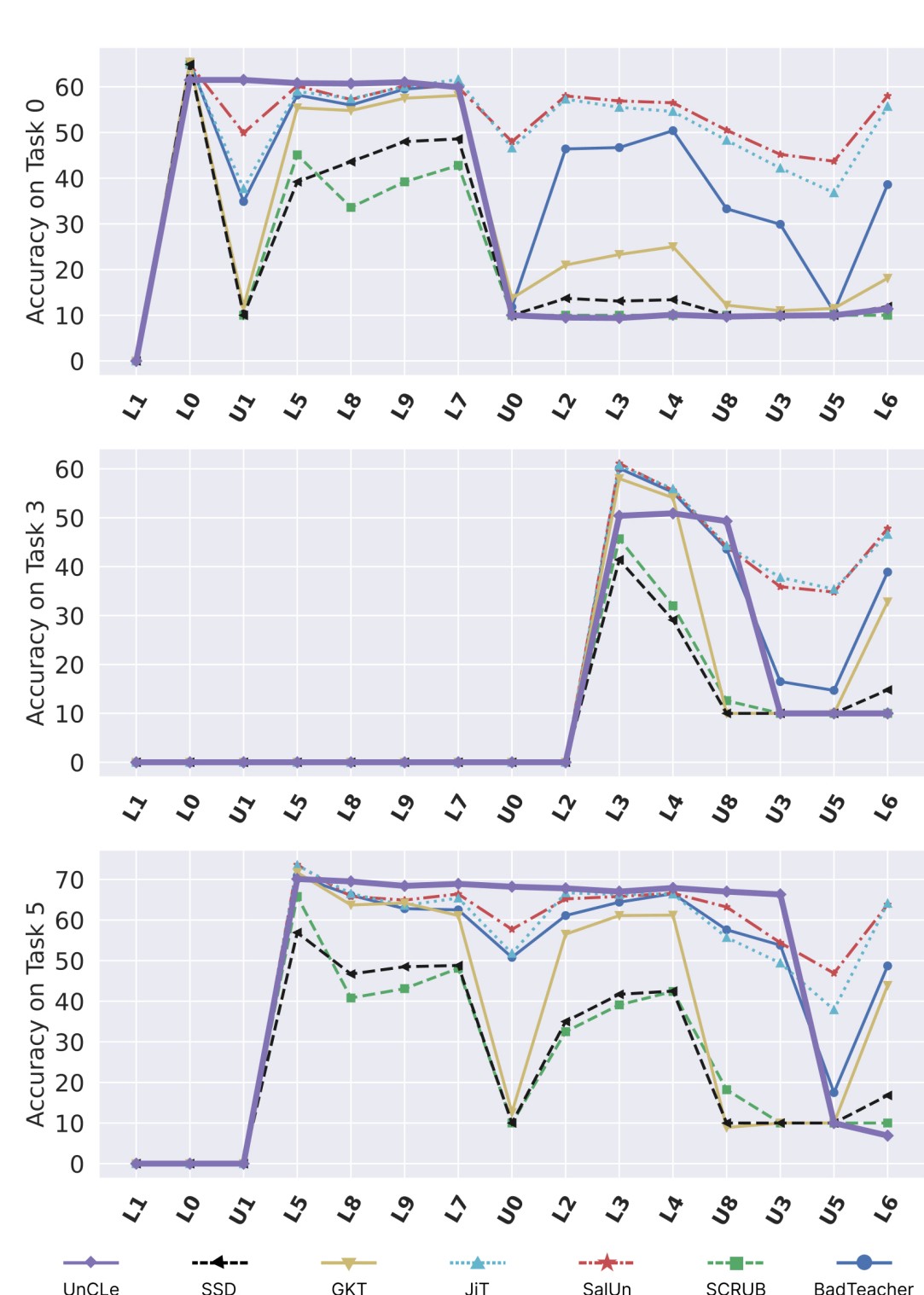

Figure E.10: Figure tracking task accuracies through the sequence of operations on the CIFAR 100 dataset. Each chart tracks a single task's accuracy as mentioned on the left.

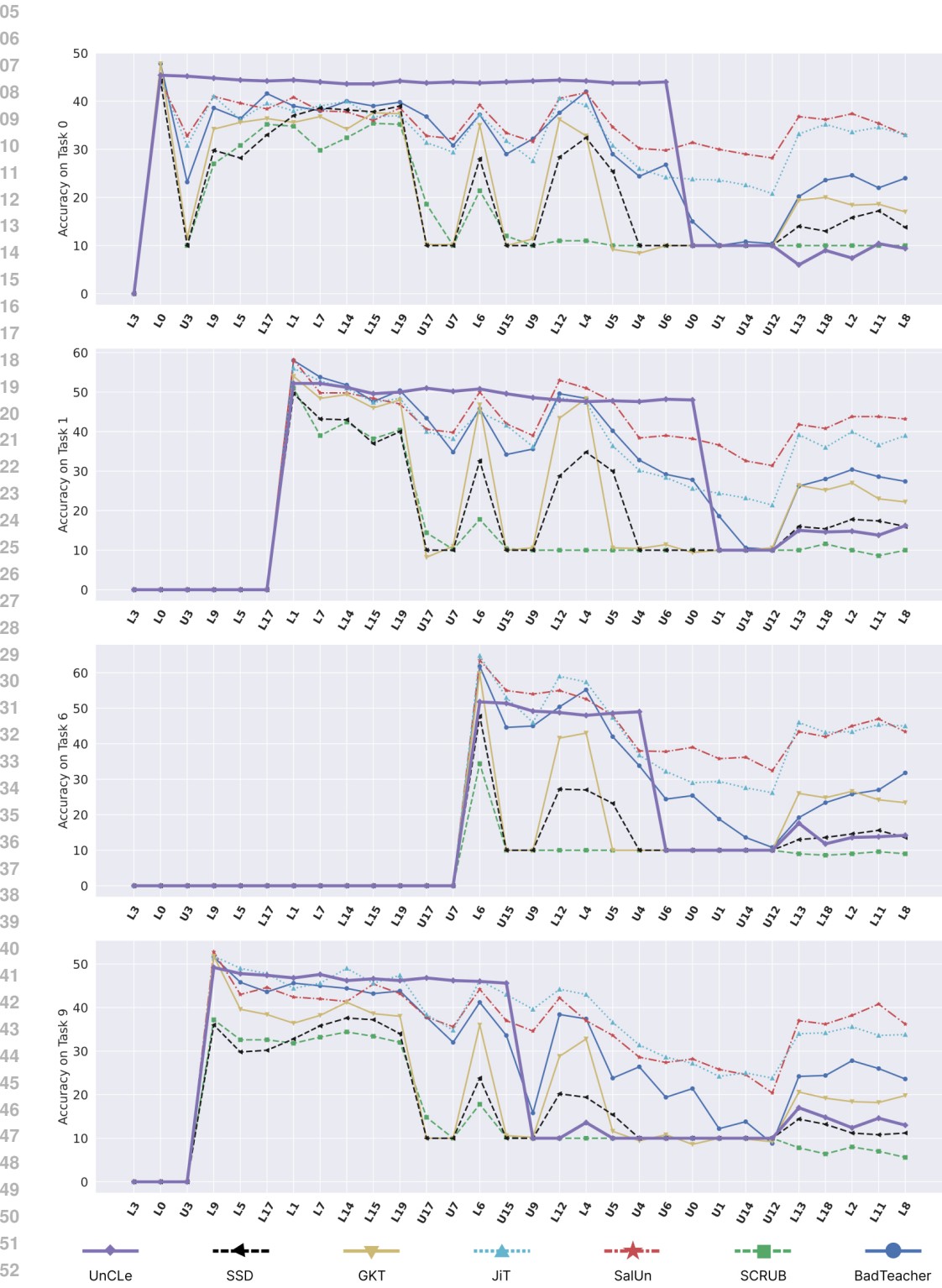

Figure E.11: Figure tracking task accuracies through the sequence of operations on the TinyImageNet dataset. Each chart tracks a single task's accuracy as mentioned on the left.

## E.5 PRIMARY EXPERIMENTS: ADDENDUM

### E.5.1 RESNET18 RESULTS

In this section, we present experiments with ResNet-18 as a backbone architecture. Each of these experiments is performed on Sequence 1 (Table D.6). The results are averaged over three runs with different seeds. We can observe from Table E.10, Table E.11, Table E.12, Table E.13, Table E.14 and Table E.15 that UnCLe performs better than all the other baselines on at least 3 out of 5 metrics. On the metric in which UnCLe is not the best, it performs equally well compared to the best one. These tables show UnCLe's superiority over other unlearning baselines.

| Methods | RA | | FA | | UNI | | SBY | | UT | |
|---|---|---|---|---|---|---|---|---|---|---|
| | mean | std | mean | std | mean | std | mean | std | mean | std |
| **BadTeacher** | 62.87 | 8.07 | 9.650 | 0.65 | 99.96 | 0.02 | 89.24 | 2.64 | 51.45 | 7.76 |
| **SCRUB** | 10.90 | 2.44 | 9.340 | 0.58 | -inf | - | 75.64 | 1.33 | 111.7 | 10.6 |
| **SalUn** | 58.94 | 9.87 | 35.16 | 5.02 | 99.35 | 0.14 | 88.95 | 2.51 | 380.9 | 30.3 |
| **JiT** | 16.66 | 2.77 | **8.990** | 1.93 | -31.09 | 42.4 | 76.87 | 1.12 | 235.8 | 56.6 |
| **GKT** | 10.82 | 1.25 | 15.21 | 1.68 | 96.37 | 0.76 | 75.35 | 0.42 | 37.39 | 6.02 |
| **SSD** | 30.22 | 22.5 | 15.07 | 6.14 | 99.99 | 0.01 | 79.74 | 5.22 | 38.46 | 3.9 |
| **Jit-Hnet** | 14.74 | 4.69 | 13.15 | 4.49 | -inf | - | 74.44 | 8.69 | 201.0 | 16.9 |
| **GKT-Hnet** | 10.07 | 0.71 | 10.69 | 1.4 | 83.19 | 1.36 | 77.10 | 0.43 | 42.92 | 2.27 |
| **UnCLe** | **93.77** | 0.40 | 9.600 | 0.99 | **100.0** | 0.00 | **99.94** | 0.04 | **10.89** | 0.02 |

Table E.10: Results on 5-Datasets (Sequence 1) with ResNet-18 Backbone

| Methods | RA | | FA | | UNI | | SBY | | UT | |
|---|---|---|---|---|---|---|---|---|---|---|
| | mean | std | mean | std | mean | std | mean | std | mean | std |
| **BadTeacher** | 65.13 | 3.67 | 10.11 | 0.52 | 99.55 | 0.01 | 88.58 | 0.4 | 6.65 | 3.29 |
| **SCRUB** | 53.39 | 3.15 | **10.00** | 0.00 | -inf | - | 74.73 | 0.26 | 18.00 | 4.14 |
| **SalUn** | **69.29** | 2.42 | 46.24 | 0.99 | 81.83 | 0.31 | 91.57 | 0.48 | 85.69 | 12.62 |
| **JiT** | 68.96 | 1.93 | 40.74 | 0.41 | 35.00 | 6.49 | 87.82 | 0.92 | 28.96 | 6.79 |
| **GKT** | 61.53 | 3.49 | 11.01 | 0.57 | 93.16 | 4.83 | 70.33 | 0.53 | 38.80 | 1.23 |
| **SSD** | 47.31 | 5.45 | **10.00** | 0.00 | 99.98 | 0.01 | 66.72 | 0.44 | **4.440** | 0.50 |
| **Jit-Hnet** | 51.52 | 18.8 | 21.84 | 4.71 | 64.49 | 14.22 | 88.21 | 4.52 | 20.50 | 1.59 |
| **GKT-Hnet** | 40.87 | 5.85 | 13.89 | 1.11 | 91.32 | 1.47 | 72.67 | 1.38 | 47.06 | 2.72 |
| **UnCLe** | 66.97 | 3.59 | **10.00** | 0.00 | **100.0** | 0.00 | **99.33** | 0.39 | 13.26 | 0.01 |

Table E.11: Results on CIFAR100 (Sequence 1) with ResNet-18 backbone

| Methods | RA | | FA | | UNI | | SBY | | UT | |
|---|---|---|---|---|---|---|---|---|---|---|
| | mean | std | mean | std | mean | std | mean | std | mean | std |
| **BadTeacher** | 53.76 | 1.63 | 12.12 | 0.52 | 99.47 | 0.02 | 85.96 | 0.12 | 6.310 | 1.28 |
| **SCRUB** | 11.71 | 1.90 | **10.00** | 0.00 | -inf | - | 70.45 | 1.35 | 17.31 | 1.09 |
| **SalUn** | 59.47 | 0.80 | 39.27 | 1.64 | 71.80 | 0.93 | 89.21 | 0.18 | 44.18 | 5.52 |
| **JiT** | **59.88** | 0.65 | 38.60 | 0.77 | 47.13 | 5.43 | 86.27 | 0.25 | 17.18 | 0.33 |
| **GKT** | 54.31 | 0.31 | 13.01 | 0.90 | 97.39 | 0.05 | 71.45 | 0.31 | 112.74 | 3.85 |
| **SSD** | 53.37 | 2.60 | 10.26 | 0.36 | 99.99 | 0.00 | 67.81 | 0.78 | **4.530** | 0.48 |
| **Jit-Hnet** | 59.20 | 1.77 | 16.32 | 0.23 | 89.57 | 2.54 | 87.32 | 1.85 | 15.37 | 0.70 |
| **GKT-Hnet** | 48.34 | 1.15 | 10.92 | 0.43 | 96.97 | 0.56 | 73.74 | 0.62 | 45.30 | 0.83 |
| **UnCLe** | 59.22 | 2.14 | **10.00** | 0.00 | **100.0** | 0.00 | **98.58** | 0.66 | 11.42 | 0.03 |

Table E.12: Results on Tiny-ImageNet (Sequence 1) with ResNet-18 backbone

| Methods | RA | | FA | | UNI | | SBY | | UT | |
|---|---|---|---|---|---|---|---|---|---|---|
| | Mean | Std | Mean | std | Mean | std | mean | std | mean | std |
| FT* | 94.47 | 0.12 | 67.70 | 2.11 | 19.93 | 0.10 | 98.52 | 1.09 | 1139 | 57.8 |
| RT* | 93.35 | 0.19 | 10.38 | 1.53 | 99.20 | 0.09 | 98.26 | 1.33 | 1532 | 436 |
| BadTeacher | 92.17 | 0.04 | 10.20 | 0.40 | 99.95 | 0.01 | 83.87 | 13.1 | 55.50 | 35.4 |
| SCRUB | 9.97 | 0.46 | **9.84** | 0.14 | -inf | - | 87.93 | 32.3 | 118.9 | 50.6 |
| SalUn | 92.39 | 0.26 | 59.24 | 2.74 | 98.47 | 0.07 | 93.53 | 4.18 | 358.3 | 51.6 |
| JiT | 86.93 | 6.09 | 29.90 | 4.96 | -3.76 | 112 | 84.52 | 13.3 | 213.7 | 44.3 |
| GKT | 89.77 | 0.31 | 12.13 | 0.95 | 96.64 | 1.32 | 72.46 | 18.1 | 36.08 | 0.07 |
| SSD | 86.32 | 0.40 | 9.93 | 0.13 | 99.66 | 0.13 | 71.88 | 18.5 | 35.16 | 14.9 |
| CLPU | 91.73 | 0.22 | **0.00** | 0.00 | - | - | 97.22 | 1.55 | **0.00** | 0.00 |
| RT-Hnet* | 70.78 | 1.71 | 14.08 | 0.54 | -30.27 | 7.54 | 78.04 | 22.6 | 1149 | 54.23 |
| Hnet | 96.60 | 0.16 | 96.91 | 0.09 | -405.1 | 19.1 | 83.59 | 15.4 | - | - |
| Jit-Hnet | 76.81 | 14.1 | 10.27 | 0.94 | 89.58 | 9.28 | 76.51 | 17.6 | 257.5 | 20.9 |
| GKT-Hnet | 95.34 | 0.37 | 14.46 | 0.35 | 91.03 | 0.55 | 75.01 | 17.6 | 43.77 | 0.34 |
| UnCLe | **96.87** | 0.20 | 10.00 | 0.06 | **100.0** | 0.00 | **99.99** | 0.01 | **13.16** | 0.05 |

Table E.13: Results on Permuted-MNIST (Sequence 1) with ResNet-18 Backbone

| Methods | RA | | FA | | UNI | | SBY | | UT | |
|---|---|---|---|---|---|---|---|---|---|---|
| | Mean | Std | Mean | std | Mean | std | mean | std | mean | std |
| FT* | 95.12 | 0.68 | 70.51 | 1.29 | 3.29 | 1.78 | 99.01 | 0.13 | 776.66 | 43.88 |
| RT* | 95.19 | 0.41 | 10.22 | 0.68 | 99.17 | 0.02 | 98.56 | 0.20 | 939.12 | 219.29 |
| BadTeacher | 94.88 | 0.30 | 9.94 | 0.54 | 99.96 | 0.00 | 90.78 | 1.97 | 50.52 | 27.79 |
| SCRUB | 10.06 | 0.07 | **9.81** | 0.31 | -inf | - | 73.16 | 0.36 | 112.72 | 40.27 |
| SalUn | 95.30 | 0.11 | 56.92 | 0.87 | 98.83 | 0.05 | 94.53 | 0.40 | 448.90 | 168.07 |
| JiT | 36.59 | 47.23 | 19.70 | 3.11 | 66.60 | 7.48 | 79.06 | 1.43 | 191.15 | 30.01 |
| GKT | 92.35 | 0.25 | 10.70 | 0.82 | 97.01 | 1.00 | 74.72 | 0.24 | 34.68 | 0.05 |
| SSD | 89.75 | 0.74 | 9.84 | 0.16 | 99.94 | 0.01 | 74.22 | 0.11 | 34.11 | 13.12 |
| CLPU | 95.21 | 0.29 | **0.00** | 0.00 | - | - | 97.72 | 0.16 | **0.00** | 0.00 |
| RT-Hnet* | 82.94 | 14.33 | 14.02 | 0.55 | -35.60 | 21.74 | 84.17 | 2.56 | 1045 | 45.6 |
| Hnet | 96.67 | 0.29 | 96.71 | 0.12 | -280.94 | 27.89 | 89.53 | 0.01 | - | - |
| Jit-Hnet | 94.15 | 2.19 | 10.55 | 0.54 | 92.11 | 7.14 | 78.65 | 2.32 | 220.32 | 44.34 |
| GKT-Hnet | 96.31 | 0.09 | 13.84 | 0.33 | 90.92 | 0.47 | 76.66 | 0.30 | 41.94 | 0.38 |
| UnCLe | **97.00** | 0.15 | 9.84 | 0.16 | **100.00** | 0.00 | **99.97** | 0.02 | **11.01** | 0.02 |

Table E.14: Results on Permuted-MNIST (Sequence 2) with ResNet-18 Backbone

| Methods | RA | | FA | | UNI | | SBY | | UT | |
|---|---|---|---|---|---|---|---|---|---|---|
| | Mean | Std | Mean | std | Mean | std | mean | std | mean | std |
| FT* | 94.22 | 0.10 | 65.17 | 1.93 | 21.65 | 1.19 | 98.40 | 0.06 | 1297 | 82.42 |
| RT* | 93.49 | 0.06 | 10.62 | 1.01 | 99.46 | 0.06 | 97.93 | 0.12 | 1505 | 412 |
| BadTeacher | 79.56 | 4.29 | 10.28 | 0.81 | 99.94 | 0.02 | 90.96 | 0.46 | 49.34 | 15.3 |
| SCRUB | 9.97 | 0.08 | 9.98 | 0.25 | -inf | - | 62.45 | 4.39 | 118.0 | 47.2 |
| SalUn | 82.40 | 0.89 | 64.78 | 2.31 | 98.30 | 0.13 | 93.50 | 0.11 | 488.8 | 203 |
| JiT | 34.45 | 42.3 | 31.00 | 11.7 | 71.94 | 11.4 | 79.05 | 10.5 | 189.0 | 36.0 |
| GKT | 12.80 | 2.35 | 11.43 | 0.72 | 96.37 | 1.44 | 68.62 | 0.19 | 36.70 | 0.07 |
| SSD | 9.90 | 0.32 | 9.92 | 0.45 | 99.99 | 0.00 | 67.81 | 0.13 | 36.81 | 9.92 |
| CLPU | 91.72 | 0.16 | **0.00** | 0.00 | - | - | 96.97 | 0.10 | **0.00** | 0.00 |
| RT-Hnet* | 49.57 | 8.69 | 16.15 | 0.74 | 5.80 | 8.45 | 69.49 | 2.50 | 1635 | 97.2 |
| Hnet | 96.80 | 0.08 | 96.72 | 0.11 | -345.1 | 29.9 | 94.59 | 0.02 | - | - |
| Jit-Hnet | 9.41 | 0.43 | **9.73** | 0.63 | -inf | - | 69.83 | 3.45 | 182.3 | 40.9 |
| GKT-Hnet | 13.96 | 2.53 | 17.25 | 2.26 | 89.65 | 0.91 | 71.46 | 0.35 | 44.55 | 0.40 |
| UnCLe | **96.98** | 0.23 | 9.93 | 0.19 | **100.0** | 0.00 | **99.99** | 0.00 | **14.79** | 0.22 |

Table E.15: Results on Permuted-MNIST (Sequence 3) with ResNet-18 Backbone

### E.5.2 RESNET50 RESULTS

The results from the primary results table, Table 1 are obtained from Sequence 1, averaged over three runs with different seeds. This section hosts the results from all three sequences, reported with mean and standard deviation obtained from averaging each experiment performed over three different seeds. The section is organized as a list of tables, with one table for each dataset-sequence pair, in the order of 5-Datasets, CIFAR-100, and Tiny-ImageNet.

| Methods | RA | | FA | | UNI | | SBY | | UT | |
|---|---|---|---|---|---|---|---|---|---|---|
| | mean | std | mean | std | mean | std | mean | std | mean | std |
| FT[*] | 88.66 | 0.45 | 67.99 | 2.83 | 23.85 | 1.66 | 97.58 | 0.15 | 1595 | 22.3 |
| RT[*] | 84.79 | 1.88 | 9.600 | 4.22 | 99.76 | 0.03 | 96.58 | 0.36 | 1566 | 19.5 |
| BadTeacher | 54.38 | 23.5 | **8.550** | 1.23 | 99.99 | 0.0 | 86.14 | 6.71 | 76.78 | 16.3 |
| SCRUB | 9.160 | 0.15 | 12.97 | 0.08 | -inf | - | 77.55 | 10.1 | 171.1 | 5.81 |
| SalUn | 74.75 | 1.56 | 25.02 | 1.22 | 99.19 | 0.02 | 93.80 | 0.27 | 491.9 | 8.01 |
| JiT | 19.10 | 13.8 | 17.20 | 3.55 | -inf | - | 87.09 | 14.8 | 242.1 | 31.4 |
| GKT | 10.27 | 0.91 | 13.67 | 1.52 | 94.58 | 2.10 | 75.24 | 0.22 | 57.67 | 5.98 |
| SSD | 8.850 | 0.00 | 10.36 | 0.09 | 99.79 | 0.05 | 72.83 | 0.40 | 47.12 | 0.45 |
| LWSF[+] | 31.76 | 0.25 | **0.00** | 0.00 | 99.98 | 0.01 | 51.21 | 1.05 | - | - |
| CLPU | 85.00 | 0.43 | **0.00** | 0.00 | - | - | 96.50 | 0.15 | **0.00** | 0.00 |
| RT-Hnet[*] | 76.23 | 3.31 | 18.44 | 0.78 | -108.5 | 71.04 | 95.63 | 0.48 | 1896 | 1.25 |
| Hnet[+] | 94.56 | 0.28 | 96.73 | 0.04 | -381.0 | 63.54 | **99.99** | 0.07 | - | - |
| Jit-Hnet | 10.19 | 1.18 | 11.29 | 4.37 | -inf | - | 73.65 | 6.20 | 306.6 | 5.08 |
| GKT-Hnet | 10.53 | 0.61 | 14.48 | 1.00 | 88.66 | 0.77 | 77.19 | 0.11 | 83.30 | 1.37 |
| UnCLe | **94.12** | 0.43 | 10.04 | 1.14 | **100.0** | 0.0 | 99.91 | 0.16 | **33.28** | 11.7 |

Table E.16: Results on 5-Datasets (Sequence 1) with ResNet-50 Backbone

| Methods | RA | | FA | | UNI | | SBY | | UT | |
|---|---|---|---|---|---|---|---|---|---|---|
| | Mean | Std | Mean | std | Mean | std | mean | std | mean | std |
| FT[*] | 88.54 | 0.53 | 58.07 | 2.4 | 42.12 | 5.03 | 95.62 | 0.13 | 3920 | 79.7 |
| RT[*] | 86.14 | 3.72 | 9.410 | 0.59 | 99.80 | 0.07 | 94.85 | 0.60 | 3851 | 68.5 |
| BadTeacher | 40.01 | 3.01 | 8.270 | 0.37 | 99.94 | 0.03 | 85.25 | 1.09 | 69.38 | 27.5 |
| SCRUB | 9.90 | 0.24 | 12.80 | 2.63 | -inf | 0 | 66.65 | 0.32 | 119.6 | 1.45 |
| SalUn | 56.29 | 7.81 | 29.40 | 2.71 | 93.56 | 1.01 | 87.62 | 1.72 | 357.5 | 4.25 |
| JiT | 11.66 | 3.51 | 22.31 | 6.3 | 19.88 | 14.3 | 77.09 | 2.48 | 170.3 | 33.6 |
| GKT | 10.52 | 0.22 | 14.44 | 0.88 | 97.24 | 0.84 | 66.98 | 0.26 | 66.48 | 11.3 |
| SSD | 10.10 | 0.01 | 14.59 | 4.66 | 100.0 | 0.0 | 66.54 | 0.68 | **33.24** | 0.19 |
| CLPU | 83.18 | 1.62 | **0.0** | 0.0 | - | - | 93.94 | 0.37 | 0.0 | 0.0 |
| RT-Hnet[*] | 62.78 | 6.57 | 10.55 | 1.01 | -75.78 | 17.3 | 85.05 | 0.04 | 3956 | 15.1 |
| Hnet[+] | **96.39** | 0.07 | 93.84 | 0.24 | -524.8 | 50.6 | **99.97** | 0.07 | - | - |
| Jit-Hnet | 9.770 | 0.23 | 17.18 | 8.8 | 75.77 | 5.97 | 83.46 | 4.39 | 202.2 | 5.89 |
| GKT-Hnet | 9.010 | 1.14 | **9.370** | 0.69 | 90.22 | 1.46 | 68.62 | 0.32 | 87.28 | 1.08 |
| UnCLe | 95.91 | 0.07 | 9.930 | 3.23 | **100.0** | 0.0 | 99.83 | 0.07 | 36.12 | 0.18 |

Table E.17: Results on 5-Datasets (Sequence 2) with ResNet-50 Backbone

| Methods | RA | | FA | | UNI | | SBY | | UT | |
|---|---|---|---|---|---|---|---|---|---|---|
| | Mean | Std | Mean | std | Mean | std | mean | std | mean | std |
| FT* | 91.21 | 0.45 | 58.63 | 0.59 | 6.520 | 2.24 | 97.75 | 0.38 | 568.0 | 15.08 |
| RT* | 91.87 | 0.66 | **7.86** | 1.81 | 99.56 | 0.06 | 95.31 | 0.42 | 551.8 | 7.12 |
| BadTeacher | 39.07 | 25.2 | 10.20 | 0.96 | 99.99 | 0.00 | 79.02 | 2.58 | 74.15 | 11.56 |
| SCRUB | 9.22 | 2.39 | 10.22 | 0.55 | -inf | - | 85.90 | 7.73 | 165.9 | 3.08 |
| SalUn | 37.55 | 6.75 | 21.99 | 1.96 | 99.22 | 0.07 | 86.75 | 0.22 | 468.0 | 6.16 |
| JiT | 12.56 | 7.53 | 11.77 | 1.43 | -55.48 | 57.9 | 73.85 | 2.30 | 225.5 | 14.94 |
| GKT | 8.35 | 0.88 | 13.03 | 1.25 | 96.71 | 0.25 | 67.29 | 0.47 | 50.69 | 0.39 |
| SSD | 12.42 | 7.55 | 10.22 | 0.55 | 99.51 | 0.78 | 73.44 | 11.7 | 46.09 | 1.24 |
| CLPU | 89.54 | 0.79 | 0.00 | 0.00 | - | - | 95.30 | 0.25 | 0.00 | 0.00 |
| RT-Hnet* | **94.05** | 0.13 | 9.350 | 0.48 | -119.1 | 68.6 | 95.89 | 1.84 | 597.2 | 12.5 |
| Hnet$^{+}$ | 92.96 | 0.13 | 93.26 | 0.08 | -442.1 | 40.1 | **99.95** | 0.05 | - | - |
| Jit-Hnet | 7.12 | 0.66 | 11.40 | 2.95 | -62.05 | 114 | 72.39 | 0.57 | 289.8 | 4.23 |
| GKT-Hnet | 15.11 | 4.94 | 13.74 | 0.90 | 91.83 | 2.07 | 72.32 | 0.64 | 76.71 | 1.85 |
| UnCLe | 93.24 | 0.76 | 11.40 | 3.05 | **100.0** | 0.00 | 99.93 | 0.07 | **19.50** | 0.00 |

Table E.18: Results on 5-Datasets (Sequence 3) with ResNet-50 Backbone

| Methods | RA | | FA | | UNI | | SBY | | UT | |
|---|---|---|---|---|---|---|---|---|---|---|
| | Mean | Std | Mean | std | Mean | std | mean | std | mean | std |
| FT* | **72.43** | 3.46 | 55.44 | 4.16 | 10.50 | 9.08 | 96.60 | 3.45 | 719.6 | 130 |
| RT* | 62.91 | 3.62 | **9.69** | 1.17 | 99.19 | 0.10 | 92.45 | 4.12 | 577.4 | 112 |
| BadTeacher | 61.75 | 4.47 | 14.57 | 0.60 | 99.63 | 0.01 | 86.13 | 6.99 | 10.95 | 2.23 |
| SCRUB | 29.45 | 7.18 | 10.06 | 0.10 | -inf | - | 64.85 | 18.9 | 30.02 | 6.96 |
| SalUn | 66.56 | 3.58 | 44.89 | 2.14 | 59.85 | 3.05 | 89.32 | 5.09 | 51.47 | 0.10 |
| JiT | 65.94 | 3.58 | 43.93 | 2.48 | 22.11 | 3.84 | 87.31 | 5.97 | 24.01 | 5.60 |
| GKT | 57.05 | 3.15 | 10.70 | 0.44 | 95.97 | 0.18 | 70.23 | 17.5 | 68.61 | 7.72 |
| SSD | 43.27 | 4.25 | 10.00 | 0.00 | 99.97 | 0.01 | 65.95 | 18.6 | **5.73** | 0.31 |
| CLPU | 63.10 | 3.77 | **0.00** | 0.00 | - | - | 91.44 | 3.93 | **0.00** | 0.00 |
| RT-Hnet* | 23.81 | 0.89 | 9.71 | 1.37 | -1.24 | 27.73 | 63.53 | 25.5 | 845.2 | 12.5 |
| Hnet | 60.52 | 3.73 | 62.84 | 2.72 | -85.50 | 25.34 | 82.74 | 15.0 | - | - |
| Jit-Hnet | 60.79 | 4.45 | 16.97 | 3.49 | 74.97 | 7.58 | 85.20 | 12.3 | 22.94 | 1.87 |
| GKT-Hnet | 40.22 | 7.49 | 9.97 | 0.83 | 90.98 | 1.43 | 73.62 | 17.9 | 83.46 | 9.58 |
| UnCLe | 62.65 | 3.85 | 10 | 0.00 | **100.0** | 0 | **99.19** | 0.42 | 41.70 | 4.25 |

Table E.19: Results on CIFAR-100 (Sequence 1) with ResNet-50 Backbone

| Methods | RA | | FA | | UNI | | SBY | | UT | |
|---|---|---|---|---|---|---|---|---|---|---|
| | Mean | Std | Mean | std | Mean | std | mean | std | mean | std |
| FT* | **73.45** | 3.47 | 57.81 | 1.24 | 9.05 | 3.15 | 97.52 | 0.17 | 387.2 | 95.2 |
| RT* | 67.42 | 2.41 | **9.84** | 1.60 | 99.14 | 0.20 | 94.03 | 0.58 | 399.5 | 61.1 |
| BadTeacher | 66.67 | 3.58 | 12.97 | 1.37 | 99.73 | 0.02 | 85.80 | 1.13 | 7.22 | 0.36 |
| SCRUB | 13.13 | 4.09 | 10.00 | 0.00 | -inf | - | 69.12 | 0.97 | 24.66 | 1.04 |
| SalUn | 72.33 | 3.00 | 44.16 | 2.21 | 53.45 | 1.56 | 90.13 | 0.53 | 46.60 | 0.32 |
| JiT | 71.80 | 3.38 | 45.98 | 0.26 | 14.26 | 3.93 | 89.21 | 0.72 | 20.15 | 1.03 |
| GKT | 61.00 | 2.27 | 11.82 | 0.85 | 95.43 | 0.64 | 72.47 | 0.22 | 61.26 | 4.15 |
| SSD | 46.45 | 1.43 | 10.00 | 0.00 | 99.56 | 0.36 | 70.55 | 0.61 | 5.38 | 0.48 |
| CLPU | 69.83 | 1.85 | **0.00** | 0.00 | - | - | 92.47 | 0.36 | 0.00 | 0.00 |
| RT-Hnet* | 44.32 | 6.60 | 10.06 | 1.06 | -9.17 | 10.1 | 72.37 | 2.86 | 412.5 | 30.8 |
| Hnet | 66.08 | 2.07 | 62.59 | 1.37 | -66.95 | 16.9 | 88.48 | 0.87 | - | - |
| Jit-Hnet | 66.97 | 2.81 | 20.24 | 2.34 | 84.11 | 3.51 | 90.76 | 4.88 | 24.10 | 6.61 |
| GKT-Hnet | 58.58 | 5.98 | 11.36 | 0.29 | 91.41 | 0.88 | 77.35 | 0.83 | 86.52 | 8.85 |
| UnCLe | 66.82 | 2.85 | 10.00 | 0.00 | **100.0** | 0.00 | **99.4** | 0.55 | **29.52** | 0.65 |

Table E.20: Results on CIFAR-100 (Sequence 2) with ResNet-50 Backbone

| Methods | RA | | FA | | UNI | | SBY | | UT | |
|---|---|---|---|---|---|---|---|---|---|---|
| | Mean | Std | Mean | std | Mean | std | mean | std | mean | std |
| FT* | **72.01** | 2.19 | 58.79 | 3.25 | 5.55 | 6.56 | 96.88 | 1.24 | 659.3 | 181 |
| RT* | 62.47 | 2.65 | 9.79 | 1.34 | 99.25 | 0.12 | 92.21 | 0.82 | 618.1 | 93.11 |
| BadTeacher | 52.76 | 1.51 | 14.55 | 1.58 | 99.56 | 0.02 | 86.65 | 0.69 | 11.47 | 1.99 |
| SCRUB | 10.00 | 0.00 | 10.00 | 0.00 | -inf | - | 61.99 | 0.78 | 32.53 | 2.85 |
| SalUn | 57.92 | 2.15 | 48.07 | 1.99 | 57.57 | 1.80 | 89.00 | 1.30 | 53.57 | 0.38 |
| JiT | 55.19 | 5.52 | 46.77 | 2.28 | 26.37 | 4.19 | 87.20 | 1.34 | 20.17 | 1.87 |
| GKT | 11.91 | 1.38 | 12.67 | 1.30 | 91.88 | 2.51 | 65.83 | 0.33 | 68.73 | 5.49 |
| SSD | 10.00 | 0.00 | 10.36 | 0.62 | 99.94 | 0.01 | 62.46 | 2.16 | **6.17** | 1.12 |
| CLPU | 61.23 | 2.56 | **0.00** | 0.00 | - | - | 90.31 | 1.71 | **0.00** | 0.00 |
| RT-Hnet* | 15.42 | 1.75 | **9.60** | 0.45 | 10.93 | 15.81 | 58.79 | 0.99 | 789.4 | 52.4 |
| Hnet | 60.66 | 2.37 | 62.04 | 0.35 | -131.33 | 25.54 | 92.77 | 0.89 | - | - |
| Jit-Hnet | 28.17 | 7.95 | 17.87 | 0.69 | 83.00 | 2.84 | 82.35 | 3.45 | 24.09 | 3.35 |
| GKT-Hnet | 9.54 | 0.94 | 11.44 | 1.49 | 89.80 | 4.72 | 67.90 | 0.34 | 93.04 | 2.41 |
| UnCLe | 58.15 | 6.09 | 10.00 | 0.00 | **100.00** | 0.00 | **98.85** | 0.74 | 41.12 | 0.59 |

Table E.21: Results on CIFAR-100 (Sequence 3) with ResNet-50 Backbone

| Methods | RA | | FA | | UNI | | SBY | | UT | |
|---|---|---|---|---|---|---|---|---|---|---|
| | Mean | Std | Mean | std | Mean | std | mean | std | mean | std |
| FT* | **60.08** | 0.30 | 52.56 | 2.38 | -11.47 | 6.08 | 95.55 | 0.30 | 694.2 | 28.6 |
| RT* | 51.86 | 0.16 | 10.47 | 0.59 | 99.23 | 0.07 | 90.74 | 0.82 | 693.2 | 27.8 |
| BadTeacher | 52.79 | 1.40 | 15.73 | 1.09 | 99.55 | 0.00 | 83.76 | 0.36 | 8.68 | 0.32 |
| SCRUB | 19.48 | 15.4 | 10.00 | 0.00 | -inf | - | 71.13 | 0.79 | 32.52 | 2.72 |
| SalUn | 58.44 | 1.57 | 36.02 | 1.23 | 65.02 | 0.70 | 86.94 | 1.14 | 65.2 | 2.15 |
| JiT | 57.86 | 2.13 | 32.70 | 0.48 | 21.10 | 4.79 | 84.42 | 0.42 | 17.71 | 0.95 |
| GKT | 52.44 | 1.53 | 11.35 | 0.77 | 97.16 | 0.75 | 70.90 | 0.54 | 147.6 | 72.8 |
| SSD | 39.78 | 3.43 | 10.37 | 0.62 | 99.98 | 0.01 | 69.70 | 1.83 | **5.81** | 0.32 |
| CLPU | 54.90 | 1.27 | **0.00** | 0.00 | - | - | 89.54 | 0.85 | **0.00** | 0.00 |
| RT-Hnet* | 53.54 | 2.76 | **9.74** | 0.86 | -23.62 | 12.3 | 73.55 | 0.40 | 758.0 | 56.0 |
| Hnet | 57.53 | 2.26 | 54.31 | 3.35 | -72.66 | 4.57 | 76.06 | 0.43 | 0.00 | 0.00 |
| Jit-Hnet | 54.10 | 2.39 | 13.05 | 0.35 | 91.07 | 1.65 | 81.61 | 0.28 | 22.83 | 3.73 |
| GKT-Hnet | 44.40 | 2.26 | 9.85 | 0.30 | 94.43 | 1.51 | 73.61 | 0.51 | 75.75 | 0.05 |
| UnCLe | 55.24 | 3.66 | 10.00 | 0.00 | **100.0** | 0.00 | **98.19** | 0.73 | 29.63 | 0.29 |

Table E.22: Results on Tiny-ImageNet (Sequence 1) with ResNet-50 Backbone

# F   COMPARISON OF REQUEST SEQUENCES

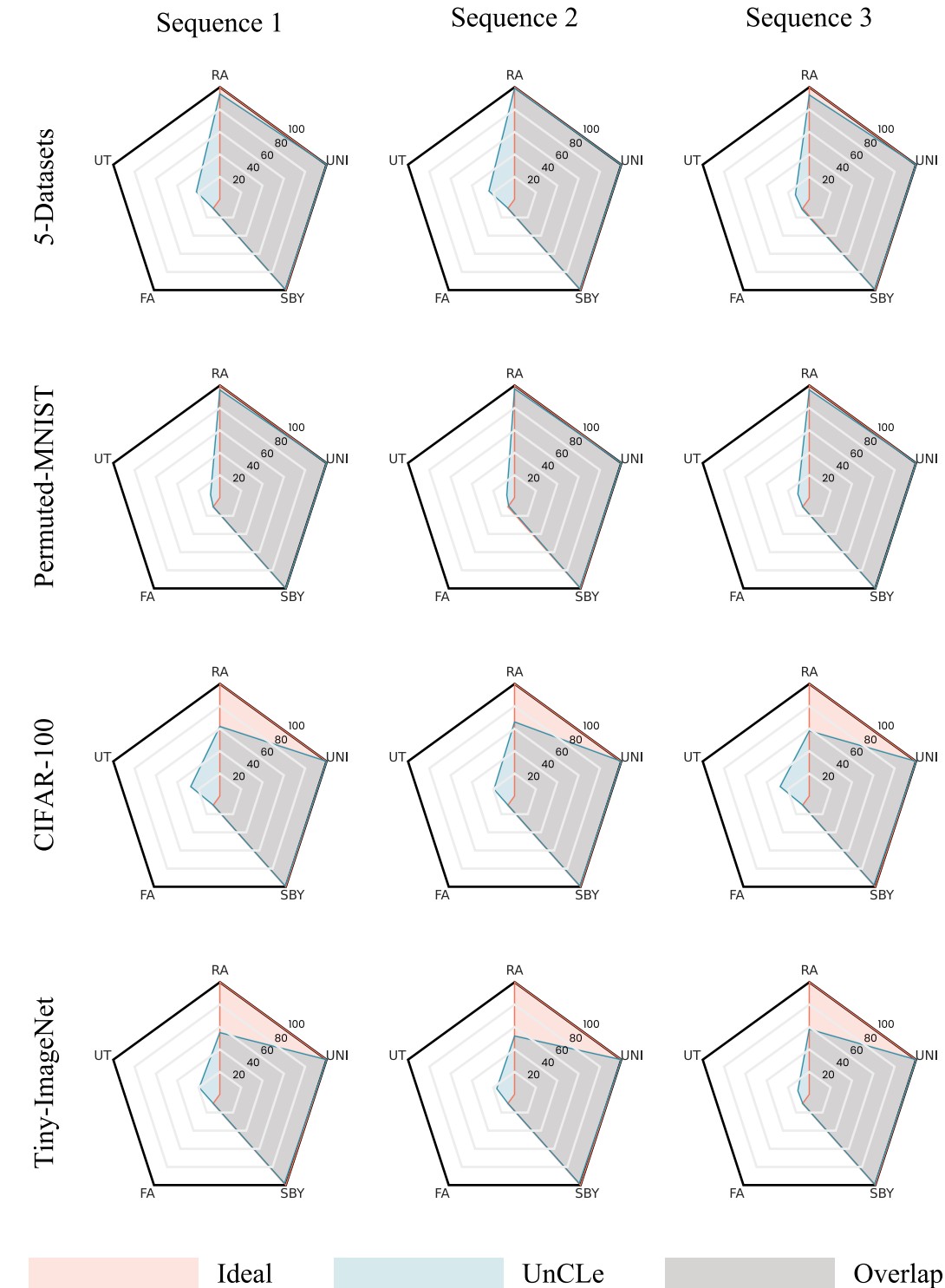

Figure F.12: A collage of radar plots displaying UnCLe's performance over different request sequences and datasets. The sequences are presented in Table D.6. This shows that UnCLe's performance is agnostic to sequences.

