# OpenReview forum: "UnCLe: An Unlearning Framework for Continual Learning"
_ICLR.cc/2025/Conference — Submitted to ICLR 2025_

### Official Review · Reviewer_9Puv · 2024-10-17

**Soundness:** 3
**Presentation:** 4
**Contribution:** 2
**Rating:** 5
**Confidence:** 4

**Summary:**

This paper studies a novel setup of continual unlearning, where tasks are repeatedly learned and unlearned in a sequence. Considering the data regulations in continual learning, the unlearning is performed without access to data. The authors propose a novel method based on hypernetwork. Specifically, the hypernetwork is trained to produce main network parameters to minimize task-specific loss for learning request, together with regularization term that retains its performance on former tasks. For unlearning request, the hypernetwork is trained to produce main network parameters that is close to Gaussian noise for unlearning target task. The proposed method outperforms existing unlearning methods in the proposed setting.

**Strengths:**

1. The paper is well-written, the methodology is clearly presented, and the readability is strong.
2. The paper considers an interesting setting where tasks can be iteratively learned and unlearned. The unlearning is conducted by data-free method, requiring no data from forget set or retain set.
3. It is demonstrated that existing method can not be directly applied for the problem setup and proposed a novel framework designed for the specific setup.

**Weaknesses:**

1. The motivation for continual unlearning requires further elaboration since this is relatively new area. In paragraph 1, machine unlearning is firstly introduced as a technique aiming to comply with privacy and regulatory requirements. However, later in this paragraph, two application examples of continual unlearning are discussed where the motivation of unlearning shifts towards utility considerations: “Unlearning obsolete skills would enable the robot to learn newer skills within the same parameter constraints”. My concerns are as follows: First, the claim that unlearning can enhance model performance on newer tasks is bold yet underexplored. Given that deep learning models exhibit sparsity, the retention of old knowledge may not significantly impede the learning of new tasks. I suggest you provide empirical evidence or citations to support the claim that unlearning can enhance performance on newer tasks. Second, the authors should clarify whether the unlearning part in continual learning aims at privacy protection or performance boost. There exists conflict mentioned above in current demonstration which may cause confusion.

2. The proposed method should align with the motivation. The unclear motivation weakens the persuasiveness of the proposed approach.. As illustrated in the paper, there is “a trivial way to unlearn” that is to discard the embedding $e_f$. But this may serve as a strong baseline. Let’s consider two circumstances: (1) Unlearning is conducted for protecting privacy. Then discarding the embedding $e_f$ or setting $\theta_f$ to be zero could suffice. To argue that the proposed method is stronger than the trivial baseline, authors should conduct experiments to demonstrate the proposed method achieves better privacy protection than these “trivial way” For example, results from MIA attacks should be reported to evaluate privacy protection, if that is indeed a key motivation. (2) Unlearning is performed for boosting performance for newer tasks. Then after unlearning the performance of later learned tasks should improve. In other words, if a task $T_f$ is unlearned at time step t, tasks learned at subsequent time steps should achieve better performance than if task $T_f$ had not been unlearned at time step t. This should be supported by empirical experiments if the motivation is indeed boosting performance. Furthermore, the aforementioned two baselines (discarding the embedding $e_f$ or setting $\theta_f$ to be zero) should be included in the experiments and compared with the proposed method.

**Questions:**

see Weaknesses

---

> ### Author Response · Authors · 2024-11-22
> **Response to Reviewer 9Puv**
>
> **We thank the reviewer for their acknowledging the presentation and soundness of the paper. We found the comments very insightful which have helped us in refining our work.**
>
> The weaknesses (W) pointed out in the comments are addressed as follows:
>
> # W1
> We apologize for the confusion in the introduction. We will edit the introduction for better clarity. The motivation behind continual unlearning is two-fold:
> * The concerns surrounding data privacy serve as our first motivation, which is the prime motivation behind machine unlearning as a whole.
> * The second motivation is unique to the continual setting wherein we show that unlearning irrelevant tasks enables better learning of newer tasks.
>
> As models continually learn more tasks, they tend to lose plasticity, and learning new tasks gets harder. At some point, the model loses all of its plasticity and doesn’t learn new tasks, as learning it would jeopardize the stability of learned tasks. This is known as the stability-plasticity dilemma [1]. The key to a successful CL model is to find the right balance between stability and plasticity. Even the best-designed models lose plasticity as they accrue more tasks. One can only fit in so much information before the model starts to catastrophically forget. Our second motivation for unlearning stems directly from this challenge. In situations where certain older tasks become obsolete, they can be unlearned. The claim that unlearning can enhance task learning can be attributed to the hypernetwork no longer being regularized to retain unlearned tasks, opening up more degrees of freedom. We explore this in further detail in the newly added **Saturation Alleviation section (E.2)** of the Appendix.
>
> # W2
> We apologize for the ambiguous presentation of the motivation and have since corrected the manuscript for greater clarity.
>
> As suggested, we consider the case wherein unlearning is conducted through discarding the task embedding. In our experiments, we find that simple discarding does not work. We’ve shown in **Figure 3 (right)** that it is possible to recover performance with a very small subset of the “unlearned” task’s data. Furthermore, as per the reviewer’s suggestions, we present the following comparison of results between discarding embeddings, setting classifier parameters to zero (Norm regularization) and UnCLe in **Table 2**. We add more details in **Baselines: Alternative Unlearning Strategies (E.1)** of the Appendix.
>
> We have already included results in **Table 3** and explanations in the results section (edited for greater clarity) that support the claims made in our second motivation that unlearning aids future task learning. The experimental setup that we’ve followed is precisely the same as the reviewer suggests. We expand **Saturation Alleviation (E.2)** of the Appendix to delve into further detail.
>
>
> ## References
> [1] Kirkpatrick, James, et al. "Overcoming catastrophic forgetting in neural networks." Proceedings of the national academy of sciences 114.13 (2017): 3521-3526.

---

> > ### Comment · Reviewer_9Puv · 2024-11-22
> > **Response to rebuttal**
> >
> > ## W1
> > The demonstration of motivation in the introduction part of the paper seems unchanged, which still raises my concern. Besides, I cannot find E.2 section and E.1 section.
> > However, the results from Table 3 indeed support the second motivation.
> >
> > ## W2
> > As for the results shown in Figure 3 (right), what are the corresponding experimental setups? How many samples are there of the whole unlearned task? The total number of samples should be stated so that readers can know the percentage of samples used for relearning. What's more, the legends in the figure should be put in order.
> > Besides, relearning with partial samples is indeed one way to evaluate unlearning methods. But it is not a typical metric. Readers may want to see more results that demonstrate the superiority of the proposed method compared to naive baselines. Table 2 seems not to present these results in the current version of the paper. And Table 2 does not agree with the description in the rebuttal response.

---

> ### Author Response · Authors · 2024-11-22
> **Paper Update**
>
> Apologies, at the time of submission of the response, the paper had not been updated. We request the reviewer to kindly check again. The motivation in the Introduction has been clarified and Table 2 has been updated as well.
>
> The experimental setup in Figure 3 (right) is as follows: A hypernetwork is trained on a multitude of Split TinyImageNet tasks with a different task embedding for each task. Trivially, a task is "unlearned" by discarding the corresponding embedding. Now, the accuracy of the unlearned task (Task 0) is measured. Then on, we consider different cases with different data subset sizes. The hypernetwork is frozen and a new task embedding is randomly initialised (given that the original embedding is discarded) and then optimized with respect to the data subset. We track the accuracy for 40 epochs which is presented in Figure 3 (right). The dashed black line (Base) shows the original performance before unlearning by discarding the embedding. This is compared with the various colored lines that each show cases with different data subset sizes used to recover the embedding.
>
> As for the number of samples in each task with regards to Figure 3 (right), there are a total of 5000 samples in each task. Our recovery results are with 10 (0.002%) samples, 20 (0.004%) samples, 50 (0.01%) samples and 100 (0.02%) samples.
>
> We measure MIA scores to demonstrate the superiority of the proposed method when compared with naive baselines. This is presented in Table 2.

---

> > ### Comment · Reviewer_9Puv · 2024-11-25
> > **Thank you for the response**
> >
> > I would like to appreciate the efforts of authors. The responses basically address my concerns, and the paper is updated accordingly.
> >
> > I have been debating whether or not to raise my score. And I finally decide to keep my score, which stems from the limited technical contribution of the paper. The proposed method is mostly based on previous work. And this is not the first research on continual unlearning. Although the experiments seem solid and the rebuttal is sufficient. I would recommend the author dive deeper into the essence of continual unlearning and enhance the technical contribution.

---

> ### Author Response · Authors · 2024-11-25
>
> We thank the reviewer for their prompt response but we respectfully disagree with the reviewer’s stance. In their own words, the reviewer did note that “the claim that unlearning can enhance model performance on newer tasks is bold yet underexplored”. We back this claim with empirical evidence and set the stage for future research to exploit this finding.
>
> Furthermore, our findings bring the permanence of existing unlearning methods into question. We find that exposing an unlearned model subsequently to similar tasks results in the model recovering its performance on the unlearned task. This defeats the whole purpose of unlearning. We also show that unlearning with UnCLe is not susceptible to such recovery attempts. We request the reviewer to kindly go through Appendix E.4 where we have shed more light upon this.
>
> While it is true that this is not the first work in Continual unlearning, our experiments demonstrate that other works are either prone to instability issues or impractical to scale. On the other hand, unlearning with UnCLe is stable, scalable and irrecoverable.
>
> We opine that the aforementioned findings are critical with palpable consequences and urgently need to be brought to the community's attention. If communicated, this work will spur further research in a number of different directions in unlearning. We request the reviewer to kindly reconsider the decision considering the work’s findings and contributions.

---

> > ### Comment · Reviewer_9Puv · 2024-11-25
> > **Response to the author**
> >
> > There may be some misunderstanding. My concern about the motivation has been resolved. My biggest concern now is the novelty of the proposed method, which seems to be a simple modification of existing methods. A more profound insight may lead to deeper investigation of continual unlearning. I certainly do not disvalue the work.

---

> > > ### Author Response · Authors · 2024-11-26
> > >
> > > The conventional approach to unlearning revolves around using data to modify model parameters. We propose an entirely new approach to unlearning in the absence of data by modifying a hypernetwork with a noise alignment objective to selectively target and purge  task-specific knowledge. Our method produces results that consistently beat existing baselines by a wide margin, all without requiring data which existing methods do.
> > >
> > > Apart from the methodology itself, our work includes findings revealing that data unlearned with existing unlearning methods are prone to recovery. The gravity of this finding necessitates communicating to the wider community. Furthermore, as the reviewer well agrees, we have shown that unlearning can enhance future learning, a finding that can potentially inspire bold new directions within both the CL and the unlearning communities. We request the reviewer to clarify the nature of the profound insights sought.

---

### Official Review · Reviewer_QS1F · 2024-11-02

**Soundness:** 3
**Presentation:** 3
**Contribution:** 3
**Rating:** 8
**Confidence:** 4

**Summary:**

The authors address the challenge of stability when unlearning in a continual learning environment. To address this challenge, the authors propose UnCLe, an Unlearning Framework for Continual Learning designed to learn tasks incrementally and unlearn tasks without access to past data. The method uses hypernetworks together with an unlearning objective that aligns task-specific parameters with noise. Experimentally they show the method’s unlearning completeness and preservation of task stability over sequences of tasks.

**Strengths:**

The research will be of interest to the ICLR community.

Originality: There is currently a lot of interest in unlearning in the continual learning setting. The authors have summarised and categorised the main approaches. They have highlighted the advantages of their approach over other approaches.

Quality: The description of the proposed method is clear and there is extensive experimentation.

Clarity: The presentation of the work is generally of high quality, and easy to read.

Experimentation: The experimentation supports the claims.

**Weaknesses:**

Significance: The significance of the work is questionable. It is incremental rather than opening a new direction of unlearning in the continual learning domain.

Motivation: The motivation given for unlearning is for privacy and the regulatory environment. The example which follows straight after this is unlearning for other reasons. I suggest that you either move the other reason for unlearning that is described in Section 6.5.1 to the motivation, or chose a motivating example that aligns with the motivation.

Claim: On page 1 there is a claim that retaining data goes against regulatory interests. In fact the opposite is true. Regulators often require data to be kept for a number of years, in case it is required as evidence.

Future work: This is missing. What are the next logical steps?

Figure 1 is a special case of unlearning where it is always unlearning the oldest task. I suggest that this is updated to the more general case where any task can be unlearnt at any time.

The grammar needs to be checked. There are prepositions missing. There are also some words that are unusual in the context in which they are used, such as "biome" on page 1, "prowess" on page 2 and "befit" on page 4.

The caption for Figure 6 has a typo: "with and with" should read "with and without"

**Questions:**

I have offered suggestions under Weaknesses above.

I am impressed with the thoroughness of the evaluation, but the paper is missing a discussion of the limitations of the approach and potential ways to address them.

What are the open problems or future directions related to your work. Adding this to the paper would improve the paper's discussion of broader impact and potential future work.

---

> ### Author Response · Authors · 2024-11-22
> **Response to Reviewer QS1F**
>
> **We thank the reviewer for their kind and encouraging words. We found their suggestions on the presentation and organization of the paper helpful. Your insights have enabled us to streamline the narrative and improve the overall quality of the work.**
>
> The weaknesses (W) pointed out in the comments are addressed as follows:
>
> # W1
> We believe that this work is significant for a number of reasons:
> * **Stability:** UnCLe is purpose-built to operate in a setting involving continual learning and unlearning. This is important because existing methods destabilize the model and cause unintentional forgetting in such dynamic settings. Our experiments show that this isn’t the case with UnCLe. (Refer Figure 4)
> * **Irrecoverability:** Unlearning with UnCLe is irrecoverable. Alarmingly, we find that this isn’t the case with existing methods, contrary to prevailing beliefs. With baseline methods, learning other tasks somehow reminds the model of the forgotten task. This spurs the need for further investigation into the reasons behind this phenomenon and methods to prevent it.
> * **Data-free unlearning:** UnCLe performs data-free unlearning using hypernetworks, making it ideal in a continual learning settings where past data are not stored.
> * **Unlearning enhances learning:** We show that unlearning obsolete tasks helps in learning new tasks better. We believe this to be a key find and a start towards more flexible CL approaches that exploit this observation.
>
> Given the multitude of observations made, we have added a **Key Takeaways (Sec 7.1)** that summarizes and highlights the paper’s key contributions. We believe that our contributions and observations are novel and set the stage for a range of diverse future research.
>
> # W2
> We’ve better motivated the introduction. We reorganized Section 6.5.1, and integrated it into Results (Section 7) in service of greater clarity. We also expand **Saturation Alleviation (E.2)** of the Appendix to delve into further detail.
>
> # W3
> Regulations are getting increasingly stringent with respect to data retention. For example, the California Privacy Rights Act (CPRA) requires companies to declare a maximum data retention period. It also emphasizes pseudonymization and anonymization, allowing companies to retain derived data such as trained models but not the original raw data if it contains personal information. In line with current trends, one can expect existing regulations to get more stringent and new privacy laws to arise across the globe.
>
> Nevertheless, we’ve removed the particular claim.
>
> # W4
> A future direction is to make UnCLe more flexible such that it can learn and unlearn individual classes (as opposed to tasks) in any arbitrary order. Another direction is to study how UnCLe can be applied to large pretrained transformer architectures and Parameter Efficient Fine Tuning methods (PEFT). We’ve incorporated limitations and future work in the Conclusion.
>
> # W5 & W6
> We apologize for the oversight. We have revised the draft and removed such errors.
>
> The questions (Q) posed in the comments are addressed as follows:
>
> # Q1
> Currently, UnCLe is capable of continually learning and unlearning tasks. In future work, we intend to make the framework more flexible by incorporating the ability to learn and unlearn individual classes. A potential approach to overcome this limitation would be to use class embeddings in place of task embeddings. These class embeddings would generate networks that learn to make a binary prediction of whether a certain data point belongs to that particular class or not. We have updated the Conclusion section to add limitations and future directions.
>
> # Q2
> As mentioned in our response to your first question, our experiments have demonstrated that there is more to unlearning and its after-effects than meets the eye. From our results, questions have surfaced as to whether the existing unlearning methods ensure an irrecoverable unlearning of data. It seems that learning similar tasks seems to “remind” models of the unlearned tasks. We consider this to be an open problem that requires further investigation to understand the reasons behind this particular phenomenon and ways apart from UnCLe in which this can be addressed.

---

> > ### Comment · Reviewer_QS1F · 2024-11-24
> > **Response to authors**
> >
> > Thanks for your feedback.

---

### Official Review · Reviewer_6wUm · 2024-11-02

**Soundness:** 2
**Presentation:** 2
**Contribution:** 2
**Rating:** 5
**Confidence:** 3

**Summary:**

The paper proposes a unified framework to consider both continuous learning and machine unlearning. Since past data is often inaccessible in some scenarios, this makes existing machine unlearning methods challenging. To address these challenges, the authors propose UnCLe, a framework for incremental task learning and unlearning in the absence of past data. UnCLe uses a hypernetwork and unlearning goals to selectively tune task-specific parameters to achieve data-free unlearning.

**Strengths:**

1 The unified continuous learning and machine unlearning framework provided by the authors is interesting and meets the challenge of not being able to access past data.

2 The proposed framework is experimented in some popular benchmarks.

3 The authors make an attempt to prevent catastrophic forgetting in the machine learning process.

**Weaknesses:**

1 Some sentences are more complex in structure and less readable, such as 'To prevent catastrophic forgetting, the hypernetwork is regularised in such a way that ...' in Chapter 5. This sentence is too long and can be broken down into two short sentences.

2  Section 5: Conversely, for an unlearning request, R t = (U, T_f , ϕ), task …’ The ϕ in ϕ and the ϕ in hypernetwork parameters ϕ are prone to ambiguity. Using different notation or adding subscripts to distinguish between the different meanings of ϕ.

3  The ResNet-50 used by the authors was too large for these tasks. The method proposed in [1] can achieve good performance in continuous learning tasks on CIFAR-10, CIFAR-100 and Tiny-ImageNet datasets (which are also used in this paper) using ResNet-18. You can justify why you did not use ResNet-18 for testing or add experiments using ResNet-18 to show the superiority of the methodology.


[1] Wang, Z., Li, Y., Shen, L., Huang, H.: A unified and general framework for continual learning. In: The Twelfth International Conference on Learning Representations (2024)

**Questions:**

1 Why must the model's prediction be similar to a random guess and close to a uniform distribution in the unlearning task? In my opinion, it probably doesn't need to be a uniform distribution to achieve unlearning capability. Could you explain why the model’s prediction must have some distributions in unlearning process.

2 In Fig.2, how to use hypernetwork to generate ResNet parameters?

---

> ### Author Response · Authors · 2024-11-22
> **Response to Reviewer 6wUm**
>
> **We thank the reviewer for their insightful comments and suggestions. We've found them helpful in enhancing the quality of our work.**
>
> The weaknesses (W) pointed out in the comments are addressed as follows:
>
> # W1 & W2
> We apologize for the ambiguity. We’ve updated the manuscript to reflect this change and have made corrections wherever required.
>
> # W3
> The hypernetwork framework can scale from a simple MLP to a large ResNet. We specifically chose a larger architecture, such as ResNet-50, to demonstrate the scalability of our approach. Nevertheless, we include additional results on ResNet-18 for clarity in **Section E.5.1 and Table E.10-12** of the Appendix.
>
> The questions (Q) posed in the comments are addressed as follows:
>
> # Q1
> Our setting is a task incremental setting where each task arrives in a sequence. Consider an exact unlearning framework that isolates each task by dedicating a separate model to learn that particular task. In such a setting, an unlearning operation would imply simply discarding the corresponding model. This is the mechanism that is employed in the Continual unlearning framework, CLPU [1].
>
> Trying to make predictions on the unlearned task for which no model exists is as good as making a random guess, where each class within the task is equally probable, given that the tasks are disjoint. This would also mean we have very high uncertainty (maximum entropy) in predicting the class. The idea is to enforce the model to learn to achieve this ideal behavior of predicting uniform distribution across classes from the unlearnt tasks.
>
> # Q2
> We have expanded the **Hypernetwork section B** of the Appendix to better explain the inner workings of the hypernetwork. We direct the reviewer’s kind attention to the same.
>
> ## References
> [1] Liu, Bo, Qiang Liu, and Peter Stone. "Continual learning and private unlearning." Conference on Lifelong Learning Agents. PMLR, 2022.

---

### Official Review · Reviewer_fr58 · 2024-11-04

**Soundness:** 3
**Presentation:** 3
**Contribution:** 2
**Rating:** 5
**Confidence:** 3

**Summary:**

The paper proposes UnCLe, an unlearning framework for continual learning designed for a sequence of learning and unlearning tasks without access to past data. UnCLe is formulated as an approximate unlearning framework.

The CL solution employs a hypernetwork that takes a task specific embedding (together with network chunk embeddings, Fig. B.7) as input to generate weights for the main network to solve a task. Furthermore, a regularization loss is used to reduce catastrophic forgetting. Sharing of hypernetwork weights between tasks leads to inter-task knowledge transfer.

The same hypernetwork is used for unlearning, where the forget task identifier is used as input (no forget data is used). The optimization objective is to reduce the MSE between the hypernetwork output and noise, averaged over n samples. Using Theorem 3 (Appendix A), the authors demonstrate that the objective effectively reduces the generated parameter values to zero, while simultaneously retaining the performance on other tasks.

The efficacy of UnCLe is demonstrated by comparing with multiple unlearning techniques, using a set of five metrics, over different datasets.

**Strengths:**

* Considering unlearning jointly within the continual learning setting is an important problem. Given that many of the unlearning techniques rely on availability of forget and retain datasets, their unavailability in this setting introduces unique challenges.

* The hypernetwork based solution is interesting.

* The authors adopt the existing unlearning methods to the CL setting for performing comparison.

**Weaknesses:**

* One major weakness in the paper is lack of insights behind empirical observations. This lack of systematic experimentation or presentation makes it difficult to assess the impact of the different solution pieces.

* Figure 3 (right) shows that one can recover the task embedding of the forgetting task. What impact does this have on the recovery of the forgetting task performance. The results of the Membership Inference attack need to be discussed more to understand the unlearning performance.

* Figure 4 (left) shows that task 0 does not suffer from catastrophic forgetting when a sequence of learning and unlearning operations are performed (x-axis). I could not find information about the dataset used for task 0, and if subsequent overlap in task data distribution is helping reduce catastrophic forgetting. How are the unlearning tasks related to the learning tasks and so on.

* The authors claim that UnCLe is agnostic to the operation sequence used. Other than the results over the three sequences in Figure F.8 is there something more concrete that can be discussed in support of this general claim.

* Other than permuted dataset, the RA scores for all the other datasets are dominated by the baseline methods. Does it show that UnCLe sacrifices learning in favor of reducing catastrophic forgetting.

* In Algorithm 2, besides the regularization term (which basically prevents the model from moving too far away from the starting values) there is nothing to prevent the unlearning from causing catastrophic forgetting. Can the authors please bring more clarity into why they observe so little impact on the existing tasks.

**Questions:**

* As per Algorithm 1, the task specific embedding is randomly initialized and then learnt as part of the optimization steps. Is this the same embedding that is then used for a forgetting task?

* The 32-dimensional task-specific embedding seem to be extremely critical for the functioning of Algorithm 1 and 2. Are there any results on the characteristics of these vectors, their alignment based on the similarity of tasks and so on.

* Thoughts on how will this hypernetwork-based solution scale to larger networks?

---

> ### Author Response · Authors · 2024-11-22
> **Response to Reviewer fr58 (Part 1/2)**
>
> **We thank the reviewer for their insightful comments which have greatly helped us refine the quality of our work.**
>
> The weaknesses (W) pointed out in the comments are addressed as follows:
>
> # W1
> In light of this, we have reorganized the Results  into a new Section (7) and provided additional insights to some of our experimental results. More importantly, we added  Key Takeaways subsection  (7.1) to better enunciate the significant insights garnered from our experiments.
>
> In line with our initial hypothesis and results presented in Table 3, our experiments showed that unlearning old tasks help learn new tasks better. We have now added  a new Saturation Alleviation section in the Appendix for a detailed understanding of this.
>
> We have clarified parts of the introduction, results and conclusion to make for a cleaner narrative. We apologize for any difficulties faced owing to lack of clarity.
>
> # W2
> The experiment behind Figure 3 (right) explores the implications of naively discarding task embeddings as a method of unlearning. As this naive method does not affect the embedding space in any way (unlike UnCLe), we show that we can recover performance on the forgotten task by attempting to relearn the discarded embedding through a very small subset of samples. This is depicted in Figure 3 (right) wherein the dashed black line shows the original performance on the forgotten task before discarding the embedding and the various coloured lines show the recovery of performance with data subsets of different sizes. To clarify, this figure is not reflective of our method UnCLe but rather an exploration of a naive unlearning method that discards the embeddings of tasks that are to be forgotten. As suggested by R4, we have included discarding of embeddings as a trivial baseline, the results of which are presented in Table 2 of the main paper and Table E.8 of the Appendix.
>
> # W3
> The dataset used in Figure 4 (left) is TinyImageNet, which is split 20 ways to create 20 different tasks with 10 classes each. The tasks are disjoint with zero overlap of classes. As observed from Figure 4 (left), it is only UnCLe that doesn’t suffer from catastrophic forgetting when unlearning operations are encountered. Baseline methods do take a hit to their performance on observing unlearning operations. Similar behavior is observed with other datasets, such as CIFAR-100, which is presented in Figure 4 (right). We apologize for the captioning error.
>
> # W4
> The general practice within the Continual Learning [1, 2, 3]  to show that a proposed approach is agnostic to the operation sequence used is to demonstrate its performance on different sequences of tasks. Following practices established in Continual Learning literature, we undertake a similar experimental setting. Nevertheless, it is known that task ordering can have an impact on the performance of CL methods, but for all practical intents and purposes, the established practice is to benchmark the algorithm on different random sequences to show that it is mostly agnostic to sequence order.
>
> ## References
> [1] Von Oswald, Johannes, et al. "Continual learning with hypernetworks." ICLR (2020)
>
> [2] Sun, Fan-Keng, Cheng-Hao Ho, and Hung-Yi Lee. "Lamol: Language modeling for lifelong language learning." ICLR (2020)
>
> [3] Chandra, Dupati Srikar, et al. "Continual learning with dependency preserving hypernetworks." WACV (2023)

---

> ### Author Response · Authors · 2024-11-22
> **Response to Reviewer fr58 (Part 2/2)**
>
> # W5
> We note that this is not a level playing field. As existing unlearning methods cannot function without data from past tasks, we endow them with a replay buffer that contains a subset of samples from each past task. Appropriately, even when learning new tasks, our baselines use data from the buffer to prevent catastrophic forgetting. Unlike our baselines, UnCLe aims to perform continual learning and unlearning in the absence of any historical data whatsoever. To alleviate catastrophic forgetting when learning and unlearning in the absence of replay, UnCLe uses regularization to preserve the generated networks pertaining to past tasks. Prior works in CL had shown that  replay-based methods can perform better than regularization-based methods, but such methods require the storage of data, which is not ideal considering the CL principle of alleviating forgetting in the absence of data.
>
> Furthermore, storing data raises strong privacy concerns that are especially pronounced in the context of Machine Unlearning. Keeping these facts in mind, we opted to pursue a regularization-based approach over replay-based approaches. Furthermore, we direct the reviewer’s attention to Figure 4 (left), which indicates how competing baselines suffer severe degradation to RA during unlearning operations, which is later rescued solely due to the presence of the replay buffer. We would like to stress the fact that in the absence of the replay buffer, the RA of the baselines would continue to degrade and would pale in comparison to UnCLe, which maintains a stable RA throughout the entire sequence of operations.
>
> # W6
> The regularization depicted in Algorithm 2 is over the output of the hypernetwork and not over the parameters of the hypernetwork itself. Whenever there is a new learning or unlearning operation, the regularization term enforces that the output of the hypernetwork in its current state is similar to that of the hypernetwork before the current operation. To do this, a copy of the hypernetwork is made, and the copy’s parameters are frozen. Now, as a new operation is performed and the hypernetwork’s parameters change, the distillation-inspired regularization term makes sure that the hypernework’s output for past tasks’ embeddings (in the case of a learning operation) or retain tasks’ embeddings (in the case of an unlearning operation) remain consistent. If the generated classifiers are largely consistent, naturally, forgetting is minimized. UnCLe is designed in such a way that both learning and unlearning operations are largely similar, barring the objective function, which holds minor changes. A more detailed explanation on continual learning with hypernetworks can be found in [1], which introduced hypernetworks for CL.
>
> The questions (Q) posed in the comments are addressed as follows:
>
> # Q1
> We use the same embedding.
>
> # Q2
> Tasks that are more similar tend to have task embeddings that are closer to each other in the embedding space. In Figure 4 of their paper, von Oswald et al [1] show that in the split MNIST benchmark (where digits are split into binary classification tasks), task-conditioned hypernetworks demonstrated alignment of embeddings based on task relationships. The embedding space revealed clusters where tasks solving similar problems (like classifying similar digits) had embeddings closer together.
>
> # Q3
> In the context of larger models such as transformers, the standard practice is to take a pre-trained model and fine-tune it continually on downstream tasks. Earlier practices involved using the pre-trained model as a starting point and fine-tuning it thereafter. Recent advances undertake parameter-efficient ways to fine-tune large models wherein the base model parameters are frozen, and a very small number of trainable task-specific parameters are added atop the existing architecture through different means to learn downstream tasks [2][3]. The hypernetwork in UnCLe can be used to generate these additional task-specific parameters, as seen in [4][5]. Adapting UnCLe to larger models and Parameter Efficient Fine-Tuning (PEFT) approaches is a future direction that we are considering.
>
> ## References
> [1] Von Oswald, Johannes, et al. "Continual learning with hypernetworks." ICLR, 2020
>
> [2] Wang, Zifeng, et al. "Learning to prompt for continual learning." CVPR. 2022.
>
> [3] B. Ermis, et al. "Continual Learning with Transformers for Image Classification," CVPRW, 2022
>
> [4] Hu, Zhiwei, et al. "HyperFormer: Enhancing entity and relation interaction for hyper-relational knowledge graph completion." Proceedings of the 32nd ACM International Conference on Information and Knowledge Management. 2023.
>
> [5] Zhang, et al. HyperPELT: Unified Parameter-Efficient Language Model Tuning for Both Language and Vision-and-Language Tasks. In Findings of the Association for Computational Linguistics: ACL 2023

---

### Author Response · Authors · 2024-11-22

**We are deeply grateful to the reviewers for their effort and time spent in reviewing our work. We value their critical and insightful comments that have greatly helped us improve the quality of our work. Furthermore, we appreciate the suggestions that have enabled us to improve the organization and the presentation of our paper.**

# Changes Made:

## Main Paper

* Introduction has been updated for greater clarity.
* Figure 1 has been corrected as per suggestions.
* Fig 4’s caption has been updated to include information about the dataset used.
* A comparison between UnCLe and alternative unlearning strategies has been added as Table 2 in place of burn in which has been moved to the Appendix.
* Results have been reorganized in the interest of clarity.
* A new section, Key Takeaways has been added to summarize the paper’s key contributions and implications.
* The Conclusions section has been updated to include limitations and future works

## Appendix:

* We have expanded the Hypernetwork section, as per R2’s requests for more detailed explanation.
* The section detailing alternative unlearning strategies has been shifted from the paper to the appendix.
* We have added a section describing Saturation Alleviation.
* The section on burn-in annealing has been moved from the paper to the appendix.
* A new section on stability with more plots has been added.
* Results on Resnet18 have been added.

---

### Meta-Review · Area_Chair_gU6G · 2024-12-18

**Metareview:**

The authors address the challenge of stability when unlearning in a continual learning environment. To address this challenge, the authors propose UnCLe, an Unlearning Framework for Continual Learning designed to learn tasks incrementally and unlearn tasks without access to past data. The method uses hypernetworks together with an unlearning objective that aligns task-specific parameters with noise. Experimentally they show the method’s unlearning completeness and preservation of task stability over sequences of tasks. There are several merits in this paper. For example, the description of the proposed method is clear and there is extensive experimentation. The presentation of the work is generally of high quality, and easy to read. The experiments support their claims. However, there are several points to be further improved. For example, the motivation for continual unlearning requires further elaboration since this is relatively new area. The significance of the work is questionable. It is incremental rather than opening a new direction of unlearning in the continual learning domain. Moreover, this paper lacks of insights behind empirical observations. This lack of systematic experimentation or presentation makes it difficult to assess the impact of the different solution pieces. Therefore, this paper cannot be accepted at ICLR this time, but the enhanced version is highly encouraged to submit other top-tier venues.

**Additional Comments On Reviewer Discussion:**

Reviewers keep the score after the rebuttal.

---

### Decision · Program_Chairs · 2025-01-22

Reject